# Debiased Contrastive Learning with multi-resolution Kolmogorov-Arnold Network for Gravitational Wave Glitch Detection

## Abstract

Time-series gravitational wave glitch detection presents significant challenges for machine learning due to the complexity of the data, limited labeled examples, and data imbalance. To address these issues, we introduce Debiased Contrastive Learning with Multi-Resolution Kolmogorov-Arnold Network(dcMltR-KAN), a novel self-supervised learning (SSL) approach that enhances glitch detection, robustness, explainability, and generalization. dcMltR-KAN consists of three key novel components: Wasserstein Debiased Contrastive Learning (wDCL), a CNN-based encoder, and a Multi-Resolution KAN (MltR-KAN). The wDCL improves the model's sensitivity to data imbalance and geometric structure. The CNN-based encoder eliminates false negatives during training, refines feature representations through similarity-based weighting (SBW), and reduces data complexity within the embedding. Additionally, MltR-KAN enhances explainability, generalization, and efficiency by adaptively learning parameters. Our model outperforms widely used baselines on O1, O2, and O3 data, demonstrating its effectiveness. Extending dcMltR-KAN to other time-series benchmarks underscores its novelty and efficiency, marking it as the first model of its kind and paving the way for future SSL and astrophysics research.

## 1 Introduction

Gravitational waves are ripples in spacetime caused by some of the most violent and energetic events in the universe, such as merging black holes, colliding neutron stars, and supernovae. Predicted by Albert Einstein in 1915 as part of his theory of general relativity, these waves carry crucial information about their origins and the nature of gravity itself. Their detection, along with the identification of glitches, offers a new way to observe the cosmos, revealing insights unattainable through traditional methods like light-based telescopes (Bi et al., 2024).

Glitches, short-lived noise transients from environmental or instrumental sources, closely mimic complex gravitational wave signals, making it difficult to distinguish real events from noise. The groundbreaking detection of gravitational waves by LIGO and Virgo in 2015 marked the beginning of a new era in astrophysics, allowing scientists to explore phenomena previously undetectable (Bailes et al., 2021). Since then, identifying and reducing gravitational wave glitches has become increasingly important in both astrophysics and deep learning (Chowdhury, 2024).

However, the unique characteristics of gravitational wave data pose significant challenges for current deep learning models, particularly in terms of data complexity, imbalance, limited labeled data, and explainability, making it difficult to accurately detect meaningful glitches. Gravitational wave raw data is inherently high-dimensional and multi-resolution, with signals captured at high sampling rates across numerous sensors and frequencies (Chua et al., 2019).

This vast, non-stationary complex data, where noise and signal properties change over time, makes it difficult for deep learning models to focus on relevant features across various scales (George & Huerta, 2018; Chowdhury, 2024). The presence of instrumental and environmental noise further complicates the distinction between real gravitational wave signals and glitches. Additionally, data imbalance—with true gravitational wave events being rare compared to the overwhelming volume of noise—leads to model overfitting on the dominant noise class, limiting detection accuracy.

Limited labels pose another obstacle to deep learning models, which typically require large amounts of labeled data for effective training. In gravitational wave glitch detection, labeled data is scarce due to the expert knowledge and resources needed for annotation (Miller & Yunes, 2019). Consequently, models often face generalization issues like underfitting or overfitting when applied to small or imbalanced datasets (Powell et al., 2023; Cuoco et al., 2020).

The lack of explainability in deep learning models is also a major limitation. As black boxes, they offer little insight into their predictions, reducing trust in gravitational wave detection where scientific validation is critical. This lack of transparency hampers collaboration with experts and complicates identifying biases or errors. These challenges highlight the shortcomings of deep learning in effectively detecting and explaining meaningful gravitational wave glitches.

To address these challenges, we introduce Debiased Contrastive Learning with Multi-Resolution Kolmogorov-Arnold Network (dcMltR-KAN), a novel self-supervised learning (SSL) framework designed to learn meaningful representations of gravitational wave data. A key component of this framework is our proposed Multi-Resolution Kolmogorov-Arnold Network (MltR-KAN), which is uniquely suited to capture multi-scale patterns and complexities inherent in gravitational wave data. Inspired by Kolmogorov's superposition theorem, MltR-KAN employs wavelet basis functions and hierarchical structures to enhance explainability, generalization, and efficiency. By leveraging the additivity and learned parameters of KAN, our approach provides interpretable insights into the detection of glitches while improving learning generalization and efficiency.

**Problem formulation.** Unlike fully-supervised deep learning approach, dcMltR-KAN offers the solution to the following problem: Given a dataset $\mathcal{X} = \{x_i\}_{i=1}^{N}$ of unlabeled gravitational wave signals, where each $x_i \in \mathbb{R}^d$ is a $d$-dimensional feature vector extracted from the raw time-series data over the period interval $T$, the goal is to learn a representation function $f : \mathbb{R}^d \to \mathbb{R}^k$ that maps signals to a $k$-dimensional feature space where meaningful signal and glitch patterns are captured.

**dcMltR-KAN** learns meaningful data representations through three novel components: Wasserstein Debiased Contrastive Learning (wDCL), a CNN-based encoder, and a multi-resolution KAN (MltR-KAN). It leverages wDCL to effectively handle both data complexity and the scarcity of labeled data, while increasing sensitivity to the geometric structure of gravitational wave glitch data and managing data imbalance. The CNN-based encoder is employed to conduct false negative elimination (FNE) in training, optimize feature representation through similarity-based weighting (SBW), and reduce data complexity via feature extraction in the embedding. MltR-KAN leverages wavelet basis functions to capture complex, multi-scale patterns in gravitational wave data. Combined with hierarchical learning, multi-resolution FNE and SBW, and learned parameters, this approach enhances glitch detection while improving efficiency, explainability, and generalization.

The proposed model surpasses other supervised deep learning approaches on the benchmark O1, O2, and O3 datasets (Abbott et al., 2019; 2021; 2023) demonstrating its superior performance and advantages. Furthermore, we extend dcMltR-KAN to benchmark audio data, showcasing its superiority. The dcMltR-KAN is the first model to integrate multi-resolution KAN into SSL, and it will inspire future research in SSL and astrophysics.

## 2 RELATED WORK

Deep learning is widely used in gravitational wave glitch detection for their capabilities in finding complex relationships and handling large-scale data.

**Generative Adversarial Networks (GANs):** Powell et al. (2023) employed GAN with advanced applications in glitch detection. Dooney et al. (2022) introduced a dual-discriminator approach for time-domain signal generation, improving convergence in gravitational wave detection. However, GANs rely on large and diverse datasets for effective training, and their performance may be limited by the availability of sufficient high-quality labeled data in the gravitational wave domain.

**Convolutional Neural Networks (CNNs):** Razzano & Cuoco (2018) developed a CNN pipeline that efficiently classified detector glitches based on their time-frequency representations, achieving high accuracy, especially in simulated data. Fernandes et al. (2023) improved glitch classification by using transfer learning with advanced CNN architectures like ConvNeXt. Alvarez-Lopez et al. (2023) integrated CNNs within a decision tree framework, showing robustness in diverse noise en-

vironments. CNNs, however, often struggle with overfitting and handling diverse noise conditions, limiting real-world generalization (Schäfer et al., 2023).

**Variational Autoencoders (VAEs) and SSL:** Sakai et al. (2022) applied a VAE with invariant information clustering (IIC) to classify transient noises by learning from 2D image features, aligning well with existing annotations. However, VAEs often produce blurry reconstructions and struggle with non-linear data, resulting in suboptimal feature representation and classification accuracy. Fernandes et al. (2023) combined SSL with CNNs to generate pseudo-labels for the Gravity Spy dataset, showing promise but falling short of supervised methods in accuracy. SSL methods often underperform fully supervised approaches, particularly in detecting subtle glitches in noisy environments.

# 3 DEBIASED CONTRASTIVE LEARNING WITH MULTI-RESOLUTION KAN

Figure 1 illustrates the proposed novel self-supervised learning (SSL) model, dcMltR-KAN, which enhances debiased contrastive learning using a multi-resolution KAN.

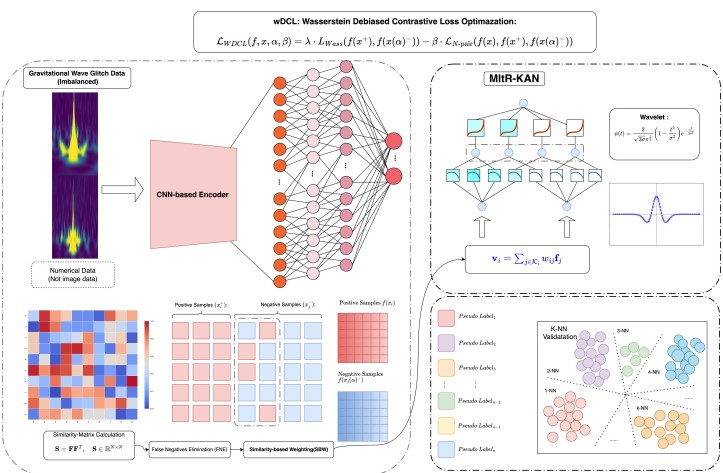

Figure 1: Debiased Contrastive Learning with Multi-Resolution Kolmogorov-Arnold Network

**dcMltR-KAN.** The model comprises three novel components. The first is the proposed Wasserstein Debiased Contrastive Learning (wDCL) that optimizes embeddings and addresses data imbalance by capturing the geometric structure of the input data.

The second is a CNN-based encoder that eliminates false negatives during training, refines feature representations in the embedding through similarity-based weighting, uncovers latent features in the input data while removing potential noise, reduces dimensionality, and lowers computational complexity in SSL.

The third component includes multi-resolution KAN (MltR-KAN) layers, which decompose the time-series glitch data into multiple resolutions using wavelets (e.g., Haar) within the KAN architecture to enhance explainability, efficiency, and generalization. Notably, SSL principles are applied across all three components: wDCL, the CNN-based encoder, and MltR-KAN. We describe each component in detail below.

## 3.1 WASSERSTEIN DEBIASED CONTRASTIVE LEARNING (wDCL)

wDCL extends debiased contrastive learning (DCL) by embedding the Wasserstein distance into the loss function, enhancing the sensitivity to the underlying geometric structure of the data and addressing data imbalance while maintaining the benefits of DCL.

**Contrastive Learning and Debiased Contrastive Learning:** Contrastive learning aims to minimize the distance between embeddings of positive sample pairs $(x, x^+)$ (similar data points), which are the transformed representations of input data after passing through an encoder, and maximize the

distance for negative sample pairs $(x, x^-)$(dissimilar data points), thereby improving the model's ability to differentiate between classes Chopra et al. (2005). Let $p_{\text{data}}(x)$ represent the data distribution of samples $x$, and let $p_{\text{pos}}(x, x^+)$ denote the distribution of positive pairs, the objective function for the encoder $f(x)$, which produces an $L_2$-normalized feature vector, is defined as: $\mathcal{L}_{\text{contrastive}} = \mathbb{E}_{(x,x^+)\sim p_{\text{pos}}}\left[-\log \frac{e^{f(x)^\top f(x^+)/\tau}}{\sum_{i=1}^M e^{f(x)^\top f(x_i^-)/\tau}}\right]$, where $x_i^-$ are negative samples drawn from $p_{\text{data}}$, $M$ is the number of negative pairs, and $\tau$ denotes the temperature parameter that controls the smoothness or sharpness of the similarity distribution in the softmax function (Wang & Isola, 2022; Chuang et al., 2020).

Contrastive learning assumes that all negative samples $x^-$ are true negatives dissimilar to the anchor sample $x$. However, in practice, some negative samples might actually be similar to the anchor (false negatives), introducing bias and degrading model performance. Debiased contrastive learning (DCL) addresses this issue by adjusting the loss function to account for the probability that a negative sample might be a false negative. The key idea is to re-weight the contribution of each negative sample based on its similarity to the anchor, thereby reducing the bias introduced by false negatives:

$$\mathcal{L}_{\text{debiased}} = \mathbb{E}_{(x,x^+)\sim p_{\text{pos}}}\left[-\log \frac{e^{f(x)^\top f(x^+)/\tau}}{e^{f(x)^\top f(x^+)/\tau} + \sum_{i=1}^M \left(e^{f(x)^\top f(x_i^-)/\tau} - \gamma e^{2f(x)^\top f(x_i^-)/\tau}\right)}\right], \text{ where } \gamma \text{ is a}$$

scaling factor representing the probability of a negative sample being a false negative.

**Wasserstein-based Debiased Contrastive Learning (wDCL):** The DCL loss function overcomes the negative pair selection limitations in constrative learning. However, the $\mathcal{L}_{\text{debiased}}$ can be sensitive to imbalanced data distributions, where negative samples may actually be similar to the anchor (false negatives). Furthermore, it can be hard for $\mathcal{L}_{\text{debiased}}$ to capture the true structure of the data, because of the limitations of Euclidean-based distances.

We propose Wasserstein-based Debiased Contrastive Learning by incorporating the Wasserstein distance into the debiased loss function to handle the challenge, forming the foundation of our dcMltR-KAN model. Unlike Euclidean or similar metrics, the Wasserstein distance captures the true geometric structure of real-world data by measuring the discrepancy between two probability distributions based on the geometry of the data space. This makes it particularly effective in high-dimensional settings. Additionally, its evaluation of entire distributions, rather than isolated points, allows it to handle imbalances between positive and negative samples more robustly. The Wasserstein distance outperforms KL-divergence by handling disjoint supports, avoiding divergence issues. It measures the "cost" of transforming one distribution into another, is robust to outliers, and ensures unbiased, symmetric comparisons, making it ideal for imbalanced datasets in contrastive learning.

The Wasserstein distance between two distributions $\mu$ and $\nu$ over a metric space $\mathcal{X}$ is defined as:

$$W(\mu, \nu) = \inf_{\gamma \in \Gamma(\mu,\nu)} \int_{\mathcal{X} \times \mathcal{X}} d(x, y) \, d\gamma(x, y), \tag{1}$$

where $\Gamma(\mu, \nu)$ denotes the set of all joint distributions $\gamma$ with marginals $\mu$ and $\nu$, and $d(x, y)$ represents the metric on space $\mathcal{X}$. Computing the Wasserstein distance is challenging due to optimization over joint distributions. We use the Sinkhorn divergence with entropy regularization for efficiency, preserving sensitivity to distributions (details in Appendix)

**Wasserstein contrastive loss $\mathcal{L}_{\text{wdcl}}$:** We can build the Wasserstein contrastive loss by integrating Wasserstein distance with $\mathcal{L}_{\text{debiased}}$. However, $\mathcal{L}_{\text{debiased}}$ may not handle multiple pairs of negatives and can not handle multiple similar and dissimilar pairs in high-dimensional spaces, besides more computing in optimization. We tackle this challenge by integrating Wasserstein distance with N-pair contrastive loss: $\mathcal{L}_{N\text{-pair}} = -\log \frac{e^{f(x)^\top f(x^+)}}{e^{f(x)^\top f(x^+)} + \sum_{i=1}^{N-1} e^{f(x)^\top f(x_i^-)}}$ as,

$$\mathcal{L}_{\text{wdcl}}(f, x, \alpha, \beta) = \lambda \cdot \mathcal{L}_{\text{wass}}(f(x^+), f(x(\alpha)^-)) - \beta \cdot \mathcal{L}_{N\text{-pair}}(f(x), f(x^+), f(x(\alpha)^-)) \tag{2}$$

Here $\mathcal{L}_{\text{wass}}(f(x^+), f(x(\alpha)^-))$ is the the Wasserstein loss capturing the geometric structure of positive and negative sample pairs, which is calculated by the Sinkhorn-divergence-based method: $L_{\text{wass}}(f(x^+), f(x^-)) = \text{Sinkhorn-divergence}(f(x^+), f(x^-))$ (Appendix). This explicit $L_{\text{wass}}$ term integrates the Wasserstein distance into the contrastive learning objective, ensuring robustness to data imbalance and alignment with the geometric structure of the feature space.

The scaling coefficients $\lambda$ and $\beta$ are employed to modulate the Wasserstein and N-pair contrastive objectives, respectively. The term $x(\alpha)^-$ refers to a subset of negative samples determined by the parameter $\alpha$. These are most likely hard negatives, selected to reduce noise and redundancy, sharpening the embedding space and improving class separability.

The hyperparameter $\alpha$ acts as a discriminative threshold to filter through negative pairs, refining the model's emphasis on important contrasts. The more details about $\alpha$ can be found in the following False Negatives Elimination (FNE) part in section 3.2. It is recommended to select $\lambda = \beta = 1$ by default, but it can be adjusted according to input data.

**Positive sample generation.** In our SSL setting, positive samples are generated by adding Gaussian noise to the input data, creating slightly varied versions of the same datapoint. This approach preserves key data characteristics while introducing variability, enhancing contrastive learning and enabling the model to better generalize across noisy and imperfect data—an advantage for glitch detection.

**Dynamic $\alpha$ adjustment**: We recommend to dynamically adjust the $\alpha$ according to input data size, which helps optimizing negative sample selection and avoid the extreme cases where those "easier" negative samples, which are already different from the anchor, are selected if a too large $\alpha$ is picked. For relatively small datasets, a smaller $\alpha$ preserves informative hard-negatives, preventing possible overfitting and enhancing generalization.

For larger datasets, a larger $\alpha$ filters out more negatives, reducing false negatives and focusing on informative hard-negatives, which are negative samples highly similar to the anchor but belong to a different class. For instance, We set from 0.1% to 4% for the O1 dataset (41,717 samples) and from 2% to 10% for larger datasets like O2 and O3 (134,372 and 500,524 samples, respectively in this study), By dynamically adjusting $\alpha$ based on the dataset size, our model adapts its learning strategy to balance easy and hard negatives, enhancing generalization and feature discrimination across task. Theorem 1 proves the robustness of wDCL to data imbalance from a loss function perspective.

**Theorem 1 (Robustness of wDCL to imbalance):** $\mathcal{L}_{\text{wdcl}}$ is robust to data imbalance than $\mathcal{L}_{\text{debiased}}$.

## 3.2 CNN-BASED ENCODER

**Rationale**: After applying wDCL, which uses the Wasserstein distance to optimize embeddings and address data imbalance by capturing the geometric structure of the data, there remains a need to refine false negative samples used in training. Furthermore, how to enhance feature representation quality for learning remains another challenge.

**CNN-based encoder.** We propose a CNN-based encoder, inspired by (Wu et al., 2018), to address these challenges using False Negatives Elimination (FNE) and Similarity-based Weighting (SBW) techniques. The encoder extracts relevant features, such as transient signal patterns or glitches, while filtering out noise in the embedding. Through convolutional, pooling, and dense layers, it captures both local and global patterns while reducing dimensionality in gravitational wave data.

**Feature similarity matrix calculation.** FNE in training and following SBW-based data representation optimization both rely on the calculation of a feature similarity matrix $\mathbf{S}$ that evaluate proximity between positive and negative pairs as follows.

The CNN-based encoder $f$ maps each input sample $x_i$ into a feature vector $\mathbf{f}_i$: $\mathbf{f}_i = f(x_i)$, $\mathbf{f}_i \in \mathbb{R}^d$, $i = 1, 2, \ldots, N$. The feature vectors $\mathbf{f}_i$ are further stacked to form the feature matrix $\mathbf{F}$:

$$\mathbf{F} = \begin{bmatrix} \mathbf{f}_1 & \mathbf{f}_2 & \cdots & \mathbf{f}_N \end{bmatrix}^\top = \begin{bmatrix} f(x_1) & f(x_2) & \cdots & f(x_N) \end{bmatrix}^\top, \quad \mathbf{F} \in \mathbb{R}^{N \times d} \tag{3}$$

We then calculate the matrix $\mathbf{S}$ to evaluate the similarity between positive and negative pairs, enabling the model to optimize the embeddings and assess how well the wDCL's learned feature space represents the underlying structure of the data under the Wasserstein contrastive loss:

$$\mathbf{S} = \mathbf{F}\mathbf{F}^T, \quad \mathbf{S} \in \mathbb{R}^{N \times N} \tag{4}$$

where each element of $\mathbf{S}$ is given by the dot product between feature vectors: $\mathbf{S}_{ij} = \mathbf{f}_i^\top \mathbf{f}_j$, where each feature vector is normalized as $\mathbf{f}_i = \frac{\mathbf{f}_i}{\|\mathbf{f}_i\|_2}$ to remove noise and possible outliers. This normalization ensures that the similarity, which is calculated as a cosine similarity, is based on the shape of the vectors rather than their magnitude, particularly important for robustly comparing noisy samples.

**False Negatives Elimination (FNE).** To enhance the quality of embeddings and eliminate false negatives during training, we utilize the matrix $\mathbf{S}$ to identify and exclude negative samples that are excessively similar to the anchor sample. For each anchor $x_i$, rank all other samples $x_j$ ($j \neq i$) by their similarity $S_{ij}$. Then we eliminate top $\alpha$ fraction by defining $\alpha \in [0, 1]$ adaptive to input data size as the elimination ratio (see dynamic $\alpha$ adjustment subsection in 3.1). With $N_{\text{neg}} = N(N - 1)$, we remove the top $\alpha \times N_{\text{neg}}$ negatives with the highest similarity scores. By removing false negatives, the refined contrastive loss $\mathcal{L}_{\text{wdcl}}$ converges faster and improves downstream task performance. Hence, FNE leads to a reduction in the overall loss, proving that it enhances SSL training quality. Mathematically, it means the following proposition holds:

**Proposition 1: $\mathcal{L}_{\text{wdcl}}$ with FNE is lower than the loss without FNE:** $\mathbb{E}_{(x,x(\alpha)^-)} \left[ \mathcal{L}_{\text{wdcl}}^{\text{FNE}} \right] < \mathbb{E}_{(x,x^-)} \left[ \mathcal{L}_{\text{wdcl}}^{\text{no FNE}} \right]$, where $\alpha$ is the the elimination ratio.

**Similarity-Based Weighting (SBW).** SBW refines feature representations in the embedding by leveraging the relationships between similar data points. We leverage the similarity matrix $\mathbf{S}$ to get $\mathbf{s}_i$, representing the similarities from sample $x_i$ to the others. We then select top $k$ similar samples by picking indices $\mathcal{K}_i$, which correspond the top $k$ highest similarity scores in $\mathbf{s}_i$. Next, we compute weights for selected indices as: $w_{ij} = \exp(s_{ij})$, $\forall j \in \mathcal{K}_i$ to emphasize the similarity and ensure positive weights. We then aggregate the top $k$ features vectors: $\mathbf{v}_i = \sum_{j \in \mathcal{K}_i} w_{ij} \mathbf{f}_j$. The aggregated feature vector $\mathbf{v}_i$ will replace the original sample $x_i$ to optimize its feature representation. This weighted aggregation improves the quality of the data for subsequent models (e.g., multiR-KAN). As such, SBW refines feature representations, leading to a lower expected contrastive loss by improving the quality of the learned embeddings. Proposition 2 demonstrates the impact of SBW on the expected wDCL loss, suggesting that the aggregated feature vector produced by SBW enhances the quality of data representation, leading to improved model performance.

**Proposition 2: The loss $\mathcal{L}_{\text{wdcl}}$ with SBW is lower than the loss without SBW**: $\mathbb{E}_{(x,\mathbf{v}_i)} \left[ \mathcal{L}_{\text{wdcl}}^{\text{SBW}} \right] < \mathbb{E}_{(x,x^-)} \left[ \mathcal{L}_{\text{wdcl}}^{\text{no SBW}} \right]$, where $\mathbf{v}_i$ is the aggregated feature vector obtained from the top $k$ most similar samples through SBW.

## 3.3 MULTI-RESOLUTION KOLMOGOROV-ARNOLD NETWORK (MLTR-KAN)

**Rationale**: While FNE and SBW refine the data representation in the embedding, they do not directly address the multi-scale patterns inherent in gravitational wave data. To handle this, we integrate a Multi-Resolution KAN, following the CNN-based encoder, into our SSL framework. MltR-KAN is a two-layer KAN with wavelet basis functions that provide built-in multi-resolution analysis and learnable parameters (Liu et al., 2024). This will enhance the explainability, efficiency, and generalization of the SSL model.

**Formulation:** Given a feature vector from the CNN-encoder $\mathbf{f}_i = f_{\text{CNN-encoder}}(x_i)$, $\mathbf{f}_i \in \mathbb{R}^n$, MltR-KAN performs the following mapping:

$$\mathbf{F}^{(l)}(\mathbf{f}_i^{(l)}) = \sum_{q=0}^{2n} \chi_q^{(l)} \left( \sum_{p=1}^{n} \psi_{pq}^{(l)}(\mathbf{f}_{ip}^{(l)}) \right), \tag{5}$$

Here, $\mathbf{F}^{(l)}(\mathbf{f}_i^{(l)})$ denotes the transformed feature vector at resolution level $l$, based on the input $\mathbf{f}_i^{(l)}$, and the number of resolution levels $L$ is set such that $2^L \leq |\mathbf{f}_i|$. $\mathbf{f}_{ip}^{(l)}$ is the $p$-th component of $\mathbf{f}_i^{(l)}$, $n$ is the total number of components (features) at each resolution level, $\psi_{pq}^{(l)}$ are the wavelet basis functions (e.g., 'db4') applied to the components of the input signal at resolution level $l$, and $\chi_q^{(l)}$ are the parameters to be learned by the MltR-KAN.

**Parameters learning.** The parameters $\chi_q$ in the mltR-KAN are learned during the SSL training using backpropagation and an optimizer (e.g., SGD). The learning process involves minimizing the Wasserstein-based debiased contrastive loss $\mathcal{L}_{\text{wdcl}}$ with respect to these parameters: $\min_{\{\chi_q\}} \mathcal{L}_{\text{wdcl}}(\mathbf{F}(\mathbf{f}_i))$. Specifically, mltR-KAN's parameters $\chi_q^{(l)}$ are updated to minimize the loss at each resolution level $l$.

**Wavelet selection.** For effective glitch detection, wavelets should be orthogonal for precise signal reconstruction, smooth with compact support to localize transient glitches, and computationally

efficient for handling large datasets. We recommend 'db4', 'sym4', or similar. The 'db4' wavelet is defined by scaling coefficients $h_n$ and wavelet coefficients $g_n = (-1)^n h_{3-n}$, with scaling function $\phi(t) = \sum_{n=0}^{3} h_n \phi(2t - n)$ and wavelet function $\psi(t) = \sum_{n=0}^{3} g_n \phi(2t - n)$. If computational speed is critical, the Haar ('db1') wavelet is orthogonal and compact but lacks smoothness. Its simplicity makes it ideal for detecting short, abrupt glitches like "blips" in gravitational wave data (Robson & Cornish, 2019).

**Enhance explainability and efficiency.** MltR-KAN leverages hierarchical learning, feature weighting, and multi-resolution FNE. MltR-KAN achieves hierarchical learning by decomposing feature representation into at multiple resolution levels $l$. It brings a hierarchical loss structure: $\mathcal{L}_{\text{wdcl}} = \sum_{l=1}^{L} \mathcal{L}^{(l)}$, where $\mathcal{L}^{(l)}$ represents the loss contribution from resolution level $l$. This decomposition enhances explainability by providing insights into how features at different resolutions contribute to the final prediction. Figure S1 in the Appendix illustrates the explainability enhancement process within the MltR-KAN model for the SNR feature extracted from the O1 data.

Moreover, feature weighting is incorporated to further refine the model's performance: $\mathcal{L}_{\text{weighted}} = \sum_{l=1}^{L} \omega^{(l)} \mathcal{L}^{(l)}$, where the learned weights $\omega^{(l)}$ adjust the contribution of each resolution level to the overall loss. These weights, distinct from the core MltR-KAN parameters (e.g., $\chi_q^{(l)}$), are optimized during training to balance multi-scale features and improve the model's efficiency and performance.

MltR-KAN reduces false negatives by leveraging multi-resolution features that capture fine details missed by single-scale models. Its hierarchical loss structure, $\mathcal{L}_{\text{FNE}} = \sum_{l=1}^{L} \mathcal{L}_{\text{FNE}}^{(l)}$, minimizes false negatives at each resolution. By focusing on fine-grained levels, the model captures subtle patterns across scales, effectively lowering the false negative rate. Similarly, we have the following result indicating that applying SBW before MltR-KAN reduces the overall loss. Figures S2 and S3 in the Appendix demonstrate a simulated training scenario under hierarchical loss where SBW or FNE is integrated within MltR-KAN. Furthermore, Proposition 3 demonstrates that applying SBW before MltR-KAN across all resolution levels reduces the overall expected wDCL loss compared to not applying SBW.

**Proposition 3: Hierarchical feature representation with Multi-Resolution SBW.** Let the overall loss be $\mathcal{L}_{\text{wdcl}} = \sum_{l=1}^{L} \mathcal{L}^{(l)}$, where $\mathcal{L}^{(l)}$ is the contribution from resolution level $l$, and $\omega^{(l)}$ are learned weights. With multi-resolution SBW applied before MltR-KAN, the refined feature representation $\mathbf{v}_i$ at each scale $l$ leads to:

$$\mathbb{E}_{x_i \sim p_{\text{data}}} \left[ \mathcal{L}_{\text{wdcl}}^{\text{SBW}} \right] < \mathbb{E}_{x_i \sim p_{\text{data}}} \left[ \mathcal{L}_{\text{wdcl}}^{\text{no SBW}} \right]. \tag{6}$$

MltR-KAN has a lower norm-based Rademacher complexity than KAN using B-spline basis functions and MLP, suggesting better generalization capability. Theorem 2 further supports that MltR-KAN offers better generalization than the standard KAN using B-spline basis functions and MLP, specifically in terms of norm-based Rademacher complexity.

**Definition: Norm-based Rademacher Complexity** of a hypothesis class $\mathcal{H}$ over a sample $S = \{x_1, \ldots, x_n\}$ is defined as: $\hat{\mathcal{R}}_n(\mathcal{H}) = \mathbb{E}_\sigma \left[ \sup_{h \in \mathcal{H}, \|h\| \leq C} \frac{1}{n} \sum_i \sigma_i h(x_i) \right]$, where $\sigma_i \in \{-1, 1\}$ are independent Rademacher variables and $\|h\| \leq C$ constrains the function norm. It quantifies the capacity of $\mathcal{H}$ to fit random noise, with lower values indicating better generalization.

**Theorem 2**: Let $\mathcal{F}_{\text{KAN-W}}$, $\mathcal{F}_{\text{KAN-S}}$, and $\mathcal{F}_{\text{MLP}}$ represent the hypothesis classes of KAN with wavelet basis, B-spline basis, and MLP, respectively. The norm-based Rademacher complexity of these classes satisfies:

$$\mathcal{R}_n(\mathcal{F}_{\text{KAN-W}}) \prec \mathcal{R}_n(\mathcal{F}_{\text{KAN-S}}) \prec \mathcal{R}_n(\mathcal{F}_{\text{MLP}}), \tag{7}$$

where $\prec$ denotes strict inequality.

**dcMltR-KAN generalization.** We have proved that the dcMltR-KAN model with wavelet basis functions exhibits a lower upper bound on the generalization error compared to dcMltR-KAN with spline basis functions or the dc-MLP model, where MltR-KAN is replaced by MLP in the proposed SSL model, as established in Theorem 2.

**Theorem 3**: Let $\mathcal{F}_{\text{dcMltR-KAN-W}}$, $\mathcal{F}_{\text{dcMltR-KAN-S}}$, and $\mathcal{F}_{\text{dc-MLP}}$ represent the hypothesis classes of dcMltR-KAN with wavelet basis, B-spline basis, and dc-MLP model, respectively. The upper-bound on the generalization error for these models satisfies:

$$\mathcal{E}_{\text{gen}}(\mathcal{F}_{\text{dcMltR-KAN-W}}) \prec \mathcal{E}_{\text{gen}}(\mathcal{F}_{\text{dcMltR-KAN-S}}) \prec \mathcal{E}_{\text{gen}}(\mathcal{F}_{\text{dc-MLP}}), \tag{8}$$

where $\mathcal{E}_{\text{gen}}(\cdot)$ denotes the generalization error, and $\prec$ signifies strict inequality.

## 4 RESULTS

We evaluate our proposed dcMltR-KAN on benchmark gravitional wave dataset (O1,O2 and O3 (Abbott et al., 2019; 2021; 2023)), besides extending it to other time-series data.

**Data and preprocessing.** We employed the benchmark O1, O2, and O3 data preprocessed from the Gravity Spy project (Glanzer et al., 2021). The preprocessed data are typically not full time-series but rather a condensed form containing key extracted features. However, challenges inherent to the original data—such as data complexity, noise, class imbalance, and the potential loss of certain temporal dynamics—can still persist Bahaadini et al. (2018).

This preprocessing derived 33 meaningful features (such as trigger timing, peak frequency, signal-to-noise ratio (SNR), amplitude, and bandwidth) from the original high-dimensional gravitational wave data, resulting in O1 with 41,717 samples and 22 glitch types, O2 with 134,372 samples and 22 types, and O3 with 500,524 samples and 24 types. Figure 2 illustrates the glitch types and their distributions for each dataset, showing that different datasets have different dominant types, with the percentage of the smallest groups reaching as low as 0.02% (e.g., Chirp in O2 and O3).

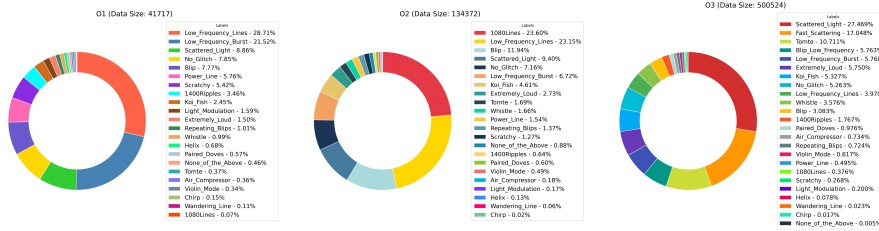

Figure 2: The imbalanced Glitch type distribution across the preprocessed O1, O2, and O3 datatsets

**Baselines:** We compare dcMltR-KAN with widely-used fully-supervised baselines (CNN, GRU, ResNet, GAN-DNN, Transformer) and three SOTA SSL models: CPC (Contrastive Predictive Coding), TS-TCC (Time-Series Representation Learning via Temporal and Contextual Contrasting), and SimCLR (Simple Contrastive Learning of Representations) (van den Oord et al., 2018; Eldele et al., 2021; Chen et al., 2020). Supervised models were trained with an 80/20 train-test split, with details in the Appendix. While CPC and TS-TCC are tailored for time-series data, SimCLR, though originally designed for other domains, has been adapted for such tasks (Zhang et al., 2022). Like its SSL peers, dcMltR-KAN is evaluated using top-1 results from a k-NN classifier on learned representations. Our implementation features a CNN-based encoder with two convolutional layers, a max-pooling layer, a dense layer, and a two-layer MltR-KAN, optimized using SGD.

**D-index.** To assess performance, we use accuracy and the D-index (Diagnostic Index) proposed by (Han et al., 2023). While accuracy can be biased in imbalanced data scenarios, the D-index effectively detects subtle performance differences and accounts for data imbalances. As an interpretable measure ranging within $(0, 2]$, the D-index measures performance by calculating the expected value of local index values across all classes:$d = \frac{1}{K} \sum_{i=1}^{K} \left( \log_2(1 + \alpha_i) + \log_2\left(1 + \frac{s_i + p_i}{2}\right) \right)$, where $\alpha_i$ is accuracy, $s_i$ is sensitivity, and $p_i$ is specificity for class $i$ among $K$ classes and $i \in K$. A higher D-index indicates better learning performance.

**Superiority of dcMltR-KAN:** Figure 3 compares dcMltR-KAN (with Haar wavelets) against the baseline models on the O1, O2, and O3 datasets. dcMltR-KAN consistently outperforms all the other models in terms of accuracy and D-index. For O1 (41,717 samples), dcMltR-KAN achieved an accuracy of $0.9817 \pm 0.0017$ and a D-index of $1.9936 \pm 0.0009$, surpassing the Transformer's accuracy of $0.9402$ and D-index of $1.9110$. This trend continues for larger datasets, such as O3 (500,524 samples), where dcMltR-KAN achieved an accuracy of $0.9009$ and a D-index of $1.9377$, outperforming the Transformer's accuracy of $0.8418$ and D-index of $1.8389$. These results demonstrate that dcMltR-KAN provides superior performance, robust generalization, and effectiveness in gravitational wave glitch detection, even compared to fully-supervised baselines.

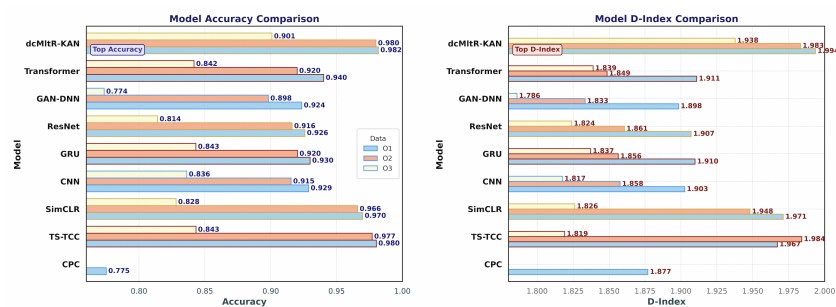

Figure 3: Comparisons of dcMltR-KAN with peer methods on O1, O2, and O3 datatsets

Although CPC showed poor performance, SimCLR and TS-TCC performed slightly worse than dcMltR-KAN for the O1 and O2 datasets in terms of accuracy and d-index. However, their performance significantly lagged behind dcMltR-KAN for the larger O3 data with 500,524 samples. This disparity underscores the superior scalability of dcMltR-KAN, which maintains robust performance even with large-scale data, thanks to its feature weighting (SBW), hierarchical learning, False Negative Elimination (FNE), and multi-resolution KAN mechanisms.

**Baseline overfitting on data imbalance.** We find that the CNN shows higher accuracy but a lower D-Index than ResNet on both the O1 and O3 datasets, suggesting it may overfit to majority classes while underperforming on minority ones. Despite CNN's higher overall accuracy (0.9288 on O1 and 0.8363 on O3), its lower D-Index reflects weaker performance on minority classes compared to ResNet (accuracy: 0.9259 on O1, 0.8141 on O3; D-Index: 1.9071 and 1.8240. In other words, while the CNN has high accuracy, it is biased toward the majority classes, meaning it overfits to the dominant patterns without effectively learning from the minority instances. Similar trends are observed for the Transformer on O2 compared to ResNet, as well as for GRU compared to the Transformer on O3. Similar trends are observed for the Transformer on O2 compared to ResNet, as well as for GRU compared to the Transformer on O3. Additionally, TS-TCC achieves 97.7% accuracy on O1 data, slightly lower than its 98.0% accuracy on O2 data. However, its D-Index on O1 is 1.984, which is higher than its D-Index on O2 (1.967), suggesting overfitting to the majority groups.

Table 1: Ablation study of dcMltR-KAN on O1, O2, and O3 data.

| Dataset | Components | Accuracy (mean $\pm$ std) | D-Index (mean $\pm$ std) |
|---------|-----------|---------------------------|--------------------------|
| O1 | *w/o wDCL* | $0.9219 \pm 0.0014$ | $1.9187 \pm 0.0015$ |
|    | *w/o mltR-KAN* | $0.9254 \pm 0.0069$ | $1.9174 \pm 0.0025$ |
| O2 | *w/o wDCL* | $0.8887 \pm 0.0015$ | $1.8154 \pm 0.0014$ |
|    | *w/o mltR-KAN* | $0.8850 \pm 0.0059$ | $1.9272 \pm 0.0035$ |
| O3 | *w/o wDCL* | $0.8888 \pm 0.0008$ | $1.9293 \pm 0.0004$ |
|    | *w/o mltR-KAN* | $0.8639 \pm 0.0018$ | $1.8830 \pm 0.0012$ |

**Abalation studies.** dcMltR-KAN consists of wDCL, a CNN-based encoder, and mltR-KAN. Since the CNN-based encoder serves as the backbone of this SSL model, we focus our ablation study on evaluating the contributions of wDCL and mltR-KAN individually. Table 1 highlights the essential roles each component plays in enhancing the model's performance across three datasets: O1, O2, and O3, where mltR-KAN with Harr. The results reveal that removing wDCL leads to a noticeable decrease in the D-Index across all datasets. Specifically, the D-Index drops to 1.9187, 1.8154, and 1.9293 on O1, O2, and O3 respectively, down from the original values of 1.9936, 1.9832, and 1.9377. Similarly, excluding mltR-KAN also results in diminished performance: on O1, the D-Index decreases to 1.9174; on O2, it decreases to 1.9272; and on O3, it decreases to 1.8830. These findings underscore the essential roles of both wDCL and mltR-KAN in maintaining and enhancing the model's performance across all evaluated datasets. The similar results can be found on other wavelets (Appendix).

**Impact of dcMltR-KAN on Data Representation** We employ UMAP to visualize gravitional wave data before and after dcMltR-KAN to examine this SSL model's impacts on data representation. Figure 4 presents UMAP visualizations of O1, O2, and O3 data, showcasing dcMltR-KAN's effec-

tiveness in enhancing feature separability and uncovering latent structures within large-scale gravitational wave data. This is further supported by silhouette analysis, which consistently validates these findings by showing a significant increase in silhouette scores for data representation after applying dcMltR-KAN (Appendix). We use UMAP instead of t-SNE because it preserves global and local structures, handles large datasets efficiently, and produces stable embeddings.

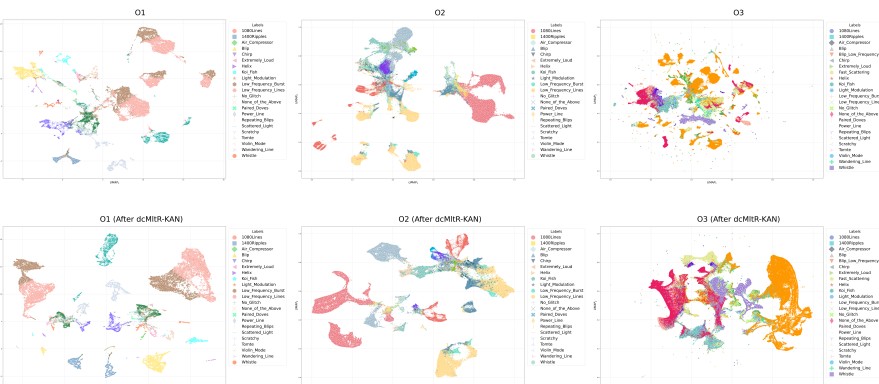

Figure 4: UMAP visualization before and after dcMltR-KAN on O1, O2 and O3 data

**Extending dcMltR-KAN to other time-series data.** We further extend dcMltR-KAN to other time-series data, demonstrating its applicability to audio tasks. For this, we use the benchmark EMODB dataset, a widely used resource for speech emotion recognition. The dataset contains 535 audio samples across seven imbalanced emotion categories: Anger (127), Boredom (81), Disgust (46), Fear (69), Happiness (71), Sadness (62), and Neutral (79). After preprocessing and feature extraction, 54 features are retained for analysis (see Appendix). dcMltR-KAN demonstrates its superiority on this dataset. Table S5 in Appendix shows the Top-1 results of dcMltR-KAN on the EMODB dataset and in the ablation study. Our model achieved 93.26% accuracy with the Mexican-hat wavelet and 88.86% accuracy with the Haar wavelet. These results outperform almost all previous fully supervised and SSL models. The ablation study further highlights the contribution of key components, showing a performance drop when wDCL or mltR-KAN is excluded. For SOTA comparison, Baek & Lee (2023) reported 90.4% weighted accuracy (WA) and 91.3% unweighted accuracy (UA) with their CNN-BiLSTM model, while Wang et al. (2023) reported 86.31% using Fairtune with a self-supervised wav2vec 2.0 model.

## 5 DISCUSSION AND CONCLUSION

While dcMltR-KAN shows strong performance in glitch detection and extends effectively to audio data, it has some weaknesses. 1) The high computational complexity of dcMltR-KAN ($\mathcal{O}(N^2+m^2)$) for large datasets like O3 ($N = 5 \times 10^6$, $m = 100$) arises from similarity matrix calculations in FNE and SBW, where $N$ and $m$ are the number of observations and training batch size during training. This challenge can be mitigated through GPU acceleration, sparsification (Liu & Liu, 2019), mini-batching (Recht et al., 2011), subsampling (Coates et al., 2011), and efficient computation libraries like cuML (Rapp et al., 2021). 2) Dynamically adjusting $\alpha$ to balance easy and hard negatives is challenging and may not generalize across datasets. Fine-tuning $\lambda$ and $\beta$ also remains complex and requires further exploration. 3) While dcMltR-KAN excels on gravitational wave and speech emotion datasets, its performance on diverse data types (e.g., audio, image) needs further evaluation.

We plan to enhance dcMltR-KAN by replacing the existing CNN-based encoder with MltR-KAN, aiming to achieve greater complexity advantages through optimized FNE and SBW directly under the MltR-KAN encoder. This will address current weaknesses and extend its applications to more data domains besides extend our model to handle raw gravitational wave data. Additionally, we intend to conduct further theoretical investigations to extend MltR-KAN in other SSL-related topics

As the first model to introduce multi-resolution KAN into SSL, dcMltR-KAN brings novelty, efficiency, and explainability to gravitational wave glitch detection and SSL, while inspiring future research in AI and astrophysics.

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

Debiased Contrastive Learning with
multi-resolution Kolmogorov-Arnold Network for
Gravitational Wave Glitch Detection **Appendix**

# 1 Wasserstein distance and Wasserstein loss $L_{\text{wass}}$ computation

The direct computation of the Wasserstein distance can be challenging due to its optimization over the space of joint distributions. We implement the Wasserstein distance more practically by using Sinkhorn divergence [4]. Sinkhorn divergence introduces entropy regularization to the optimal transport problem, making it efficient to compute while retaining sensitivity to distributional characteristics.

Given two distributions $\mu$ and $\nu$, with corresponding samples $\{x_i\}_{i=1}^n$ and $\{y_j\}_{j=1}^m$, the Sinkhorn divergence is defined as:

$$W_\epsilon(\mu, \nu) = \min_{\gamma \in \Gamma(\mu,\nu)} \sum_{i,j} \gamma_{ij} c(x_i, y_j) + \epsilon \cdot \text{KL}(\gamma \| \mu \otimes \nu), \tag{1}$$

where:

- $\Gamma(\mu,\nu)$: Set of all joint distributions with marginals $\mu$ and $\nu$.

- $\gamma_{ij}$: Transport plan between $x_i$ and $y_j$.

- $c(x_i, y_j)$: Cost function, often the squared Euclidean distance $\|x_i - y_j\|^2$.

- $\epsilon$: Regularization parameter for entropy smoothing.

- $\text{KL}(\gamma \| \mu \otimes \nu)$: Kullback-Leibler divergence regularizing the transport plan.

- $\mu \otimes \nu$: Independent product of the distributions $\mu$ and $\nu$.

## 1.1 Wasserstein loss $L_{\text{wass}}(f(x^+), f(x^-)) = \textbf{Sinkhorn-divergence}(f(x^+), f(x^-))$

Our Wasserstein loss $L_{\text{wass}}$ is defined as

$$L_{\text{wass}}(f(x^+), f(x^-)) = \min_{\gamma \in \Gamma(f(x^+),f(x^-))} \sum_{i,j} \gamma_{ij} c(f(x_i^+), f(x_j^-)) + \epsilon \cdot \text{KL}(\gamma \| f(x^+) \otimes f(x^-)), \tag{2}$$

where:

- $\Gamma(f(x^+), f(x^-))$: Set of all joint distributions between the embeddings of positive samples $f(x^+)$ and negative samples $f(x^-)$.

- $\gamma_{ij}$: Transport plan between $f(x_i^+)$ and $f(x_j^-)$.

- Cost function $c(f(x_i^+), f(x_j^-))$: $\frac{1}{2}\|f(x_i^+) - f(x_j^-)\|^2$.

- $\epsilon$ (0.01): Regularization parameter for entropy smoothing .

- $\text{KL}(\gamma \| f(x^+) \otimes f(x^-))$: Kullback-Leibler divergence regularizing the transport plan.

- $f(x^+) \otimes f(x^-)$: Independent product of the embeddings' distributions.

# 2  Proof of Theorem 1.

## Theorem 1 (Robustness of wDCL to Data Imbalance):

$\mathcal{L}_{\text{wdcl}}$ is robust to data imbalance than $\mathcal{L}_{\text{debiased}}$.

*Proof.*

Let $\mathcal{L}_{\text{wdcl}}$ represent the Wasserstein-based Debiased Contrastive Loss, and $\mathcal{L}_{\text{debiased}}$ represent the standard Debiased Contrastive Loss, we will demonstrate that the Wasserstein distance term in $\mathcal{L}_{\text{wdcl}}$ provides a more stable and representative measure of dissimilarity between distributions, especially under data imbalance.

### 1. Sensitivity of $\mathcal{L}_{\text{debiased}}$ to Data Imbalance:

The Debiased Contrastive Loss $\mathcal{L}_{\text{debiased}}$ is defined as:

$$
\mathcal{L}_{\text{debiased}} = \mathbb{E}_{(x,x^+)\sim p_{\text{pos}}} \left[ -\log \frac{e^{f(x)^\top f(x^+)/\tau}}{e^{f(x)^\top f(x^+)/\tau} + \sum_{i=1}^{M}\left(e^{f(x)^\top f(x_i^-)/\tau} - \gamma e^{2f(x)^\top f(x_i^-)/\tau}\right)} \right]
\tag{3}
$$

This loss function relies on pointwise similarities between embeddings $f(x_i^-)$ and $f(x)$. Under data imbalance, where negative samples $x_i^-$ dominate, the pointwise similarities become biased, resulting in gradient updates that do not reflect the true data structure!

It means that the variance of the gradient updates under data imbalance becomes higher: $\text{Var}(\nabla_f \mathcal{L}_{\text{debiased}})$ is large due to this overrepresentation.

### 2. Robustness of the Wasserstein Distance:

The Wasserstein distance $W(\mu, \nu)$ between two probability distributions $\mu$ and $\nu$ over a metric space $\mathcal{X}$ is defined as:

$$
W(\mu,\nu) = \inf_{\gamma \in \Gamma(\mu,\nu)} \int_{\mathcal{X}\times\mathcal{X}} d(x,y)\, d\gamma(x,y),
\tag{4}
$$

where $\Gamma(\mu, \nu)$ is the set of all couplings (joint distributions) with marginals $\mu$ and $\nu$, and $d(x, y)$ is a distance metric.

*a. Sensitivity to Distribution Geometry:* The Wasserstein distance captures global structural differences by considering optimal mass transport between distributions, not pointwise similarities.

*b. Robustness to Data Imbalance:* The Wasserstein distance evaluates the entire distribution's transport plan, making it less influenced by sample imbalance and mitigating negative sample overrepresentation.

### 3. Wasserstein-based Debiased Contrastive Loss $\mathcal{L}_{\mathbf{wdcl}}$:

The Wasserstein-based Debiased Contrastive Loss is defined as:

$$\mathcal{L}_{\mathrm{wdcl}}(f, x, \alpha, \beta) = \lambda \mathcal{L}_{\mathrm{wass}}(f(x^+), f(x(\alpha)^-)) - \beta \mathcal{L}_{\mathrm{N\text{-}pair}}(f(x), f(x^+), f(x(\alpha)^-)), \tag{5}$$

where $\mathcal{L}_{\mathrm{wass}}$ represents the Wasserstein distance between positive and negative samples.

### 4. Gradient Stability and Generalization Comparison:

**Gradient Stability:** The gradient for $\mathcal{L}_{\mathrm{wdcl}}$ with respect to the model parameters $f$ is given by:

$$\nabla_f \mathcal{L}_{\mathrm{wdcl}} = \lambda \nabla_f \mathcal{L}_{\mathrm{wass}} - \beta \nabla_f \mathcal{L}_{\mathrm{N\text{-}pair}}. \tag{6}$$

The Wasserstein term involves integration over the distributions, leading to smoother gradients:

$$\mathrm{Var}(\nabla_f \mathcal{L}_{\mathrm{wdcl}}) < \mathrm{Var}(\nabla_f \mathcal{L}_{\mathrm{debiased}}). \tag{7}$$

**Generalization:** Generalization error is given by the expected difference between the true data distribution $P_{\mathrm{data}}$ and the model's learned distribution $Q_{\mathrm{model}}$:

$$\mathbb{E}_{x \sim P_{\mathrm{data}}}[\mathcal{L}(f(x))] - \mathbb{E}_{x \sim Q_{\mathrm{model}}}[\mathcal{L}(f(x))]. \tag{8}$$

For $\mathcal{L}_{\mathrm{wdcl}}$, this difference is minimized, as it reflects the global structure of the data.

### 5. Mathematical Justification:

**For $\mathcal{L}_{\mathbf{debiased}}$:** The gradient with respect to $f(x)$ is influenced by individual negative samples $f(x_i^-)$. Overrepresentation of negative samples leads to biased gradient updates.

**For $\mathcal{L}_{\mathbf{wdcl}}$:** The Wasserstein term involves integration over distributions:

$$\nabla_f \mathcal{L}_{\mathrm{wass}} \propto \int_{\mathcal{X}} \left( \nabla_f f(x^+) - \nabla_f f(x_i^-) \right) d\gamma(x^+, x_i^-), \tag{9}$$

leading to smoother gradients and less sensitivity to imbalance.

By incorporating the Wasserstein distance, $\mathcal{L}_{\mathrm{wdcl}}$ smooths the effect of imbalanced samples and better captures the global structure of the data, resulting in:

- **More Stable Optimization:** Gradients are less volatile:

$$\mathrm{Var}(\nabla_f \mathcal{L}_{\mathrm{wdcl}}) < \mathrm{Var}(\nabla_f \mathcal{L}_{\mathrm{debiased}}). \tag{10}$$

- **Better Generalization:** The model learns embeddings that reflect the true data distribution:

$$\mathbb{E}_{x \sim P_{\text{data}}}[\mathcal{L}(f(x))] - \mathbb{E}_{x \sim Q_{\text{model}}}[\mathcal{L}(f(x))] \quad \text{is minimized for} \quad \mathcal{L}_{\text{wdcl}}. \quad (11)$$

Therefore, mathematically, $\mathcal{L}_{\text{wdcl}}$ is more robust to data imbalance than $\mathcal{L}_{\text{debiased}}$.

# 3 Proof of Theorem 2

**Theorem 2** Let $\mathcal{F}_{\text{KAN-W}}$, $\mathcal{F}_{\text{KAN-S}}$, and $\mathcal{F}_{\text{MLP}}$ be the hypothesis classes of KAN with wavelet basis functions, B-spline basis functions, and MLP (Multilayer Perceptron), respectively. The norm-based Rademacher complexity of these function classes satisfies the following inequality:

$$\mathcal{R}_n(\mathcal{F}_{\text{KAN-W}}) \prec \mathcal{R}_n(\mathcal{F}_{\text{KAN-S}}) \prec \mathcal{R}_n(\mathcal{F}_{\text{MLP}}), \tag{12}$$

, where $\prec$ denotes a strict inequality.

   *Proof.*

   We begin with the empirical Rademacher complexity for a function class $\mathcal{F}$ over a sample $S = \{x_1, \ldots, x_n\}$:

$$\mathcal{R}_n(\mathcal{F}) = \mathbb{E}_\sigma \left[ \sup_{f \in \mathcal{F}} \frac{1}{n} \sum_{i=1}^n \sigma_i f(x_i) \right], \tag{13}$$

where $\sigma_i$ are Rademacher random variables taking values in $\{-1, 1\}$ with equal probability, and $f(x_i) \in \mathcal{F}$ represents the function applied to sample $x_i$.

   The Rademacher complexity can be bounded based on the norm of the hypothesis class, using the inequality:

$$\mathcal{R}_n(\mathcal{F}_\psi) \leq \frac{\lambda}{\sqrt{n}} \cdot \mathbb{E} \left[ \|\psi\|_{\mathcal{H}_\psi} \right], \tag{14}$$

   where $\|\psi\|_{\mathcal{H}_\psi}$ is the norm of the basis function $\psi$ in the appropriate Hilbert space $\mathcal{H}_\psi$, and $\lambda$ is a constant.

   We compare the norms of the basis functions:

   1. **Wavelet Basis** $\psi_w$: Wavelet functions have compact support and exhibit localization in both time and frequency. The $H^1$ norm of a wavelet basis function is given by:

$$\|\psi_w\|_{H^1} = \int_{-\infty}^{\infty} \left( |\psi_w(x)|^2 + |\nabla \psi_w(x)|^2 \right) dx.$$

Since wavelets are localized, this norm is relatively small, leading to a lower Rademacher complexity.

   2. **B-Spline Basis** $\psi_s$: Spline basis functions are smoother but more global than wavelets. Their $H^1$ norm is given by:

$$\|\psi_s\|_{H^1} = \int_0^1 \left( |\psi_s(x)|^2 + |\nabla \psi_s(x)|^2 \right) dx. \tag{15}$$

   B-Splines typically have larger norms because they spread over larger intervals and require more parameters, leading to a higher complexity compared to wavelets.

   3. **MLP Functions**: MLPs, with many parameters, exhibit high expressivity but also have very large norms due to the number of layers and parameters. Therefore, the Rademacher complexity of MLPs grows significantly faster than that of wavelet and spline functions.

To rigorously quantify these differences, we apply Dudley's entropy integral:

$$\mathcal{R}_n(\mathcal{F}) \leq \frac{12}{\sqrt{n}} \int_0^\infty \sqrt{\log N(\epsilon, \mathcal{F}, \|\cdot\|)} \, d\epsilon, \tag{16}$$

where $N(\epsilon, \mathcal{F}, \|\cdot\|)$ is the covering number of $\mathcal{F}$ with $\epsilon$-balls under the norm $\|\cdot\|$. Since wavelets require fewer terms to represent functions, the covering number is smaller for $\mathcal{F}_{\text{KAN-W}}$, followed by $\mathcal{F}_{\text{KAN-S}}$, and then $\mathcal{F}_{\text{MLP}}$.

Thus, integrating the bounds gives:

$$\mathcal{R}_n(\mathcal{F}_{\text{KAN-W}}) \leq \frac{12}{\sqrt{n}} \int_0^\infty \sqrt{\log N(\epsilon, \mathcal{F}_{\text{KAN-W}}, \|\cdot\|)} \, d\epsilon, \tag{17}$$

with the same inequality holding for $\mathcal{F}_{\text{KAN-S}}$ and $\mathcal{F}_{\text{MLP}}$.

Therefore, by combining norm-based bounds and entropy integrals, we conclude:

$$\mathcal{R}_n(\mathcal{F}_{\text{KAN-W}}) \prec \mathcal{R}_n(\mathcal{F}_{\text{KAN-S}}) \prec \mathcal{R}_n(\mathcal{F}_{\text{MLP}}) \tag{18}$$

# 4 Proof of Theorem 3

**Theorem 3**: Let $\mathcal{F}_{\text{dcMltR-KAN-W}}$, $\mathcal{F}_{\text{dcMltR-KAN-S}}$, and $\mathcal{F}_{\text{dc-MLP}}$ represent the hypothesis classes of dcMltR-KAN with wavelet basis, B-spline basis, and dc-MLP model, respectively. The upper-bound on the generalization error for these models satisfies:

$$\mathcal{E}_{\text{gen}}(\mathcal{F}_{\text{dcMltR-KAN-W}}) \prec \mathcal{E}_{\text{gen}}(\mathcal{F}_{\text{dcMltR-KAN-S}}) \prec \mathcal{E}_{\text{gen}}(\mathcal{F}_{\text{dc-MLP}}), \qquad (19)$$

where $\mathcal{E}_{\text{gen}}(\cdot)$ denotes the generalization error, and $\prec$ signifies strict inequality.

*Proof.*

## 1. Preliminaries

We are to prove that the upper bound on the generalization error for the models satisfies: $\mathcal{E}_{\text{gen}}(\mathcal{F}_{\text{dcMltR-KAN-W}}) \prec \mathcal{E}_{\text{gen}}(\mathcal{F}_{\text{dcMltR-KAN-S}}) \prec \mathcal{E}_{\text{gen}}(\mathcal{F}_{\text{dc-MLP}})$,

where:

- $\mathcal{F}_{\text{dcMltR-KAN-W}}$ is the hypothesis class of the dcMltR-KAN model with wavelet basis functions.

- $\mathcal{F}_{\text{dcMltR-KAN-S}}$ is the hypothesis class of the dcMltR-KAN model with B-spline basis functions.

- $\mathcal{F}_{\text{dc-MLP}}$ is the hypothesis class where MltR-KAN is replaced by an MLP.

The generalization error $\mathcal{E}_{\text{gen}}(\mathcal{F})$ measures the difference between the expected loss and the empirical loss for a hypothesis class $\mathcal{F}$:

$$\mathcal{E}_{\text{gen}}(\mathcal{F}) = \mathbb{E}_{f \sim \mathcal{F}}[L_{\text{expected}}(f) - L_{\text{empirical}}(f)]. \qquad (20)$$

The Rademacher complexity $\mathcal{R}_n(\mathcal{F})$ of a hypothesis class $\mathcal{F}$ with sample size $n$ is a measure of its capacity, reflecting how well the class can fit random noise:

$$\mathcal{R}_n(\mathcal{F}) = \mathbb{E}_{\sigma, X} \left[ \sup_{f \in \mathcal{F}} \frac{1}{n} \sum_{i=1}^{n} \sigma_i f(x_i) \right], \qquad (21)$$

where $\sigma_i$ are independent Rademacher variables taking values $\pm 1$ with equal probability, and $X = \{x_1, \ldots, x_n\}$ is the sample.

## 2. Relate generalization error to Rademacher complexity

We can have the relationships between the generalization error and the Rademacher complexity:

$$\mathcal{E}_{\text{gen}}(\mathcal{F}) \leq 2\mathcal{R}_n(\mathcal{F}) + \epsilon(n, \delta), \qquad (22)$$

where $\epsilon(n, \delta)$ is a term that diminishes as the sample size $n$ increases and confidence level $\delta$ is considered. Since $\epsilon(n, \delta)$ is common for all models (assuming the same $n$ and $\delta$), the primary factor influencing the generalization error is the Rademacher complexity $\mathcal{R}_n(\mathcal{F})$.

### 3. Apply Theorem 2

From Theorem 2, we have the ordering of Rademacher complexities:

$$\mathcal{R}_n(\mathcal{F}_{\text{dcMltR-KAN-W}}) \prec \mathcal{R}_n(\mathcal{F}_{\text{dcMltR-KAN-S}}) \prec \mathcal{R}_n(\mathcal{F}_{\text{dc-MLP}}). \qquad (23)$$

Since the generalization error is directly proportional to the Rademacher complexity, we have the ordering of generalization errors follows the same strict inequalities:

$$\mathcal{E}_{\text{gen}}(\mathcal{F}_{\text{dcMltR-KAN-W}}) \prec \mathcal{E}_{\text{gen}}(\mathcal{F}_{\text{dcMltR-KAN-S}}) \prec \mathcal{E}_{\text{gen}}(\mathcal{F}_{\text{dc-MLP}}). \qquad (24)$$

This result indicates that the dcMltR-KAN model with wavelet basis functions has a strictly lower upper bound on the generalization error compared to the versions with B-spline basis. This is because Wavelet Basis Functions offer a sparse representation and capture localized features effectively, leading to a more constrained hypothesis class with lower complexity.

# 5 Ablation studies of dcMltR-KAN with four wavelets

Table 1: Ablation Study: Top-1 Accuracy and D-Index for Different Methods (Datasets: O1, O2, and O3)

| Dataset | Method | Top-1 Accuracy (mean ± std) | D-Index (mean ± std) |
|---------|--------|------------------------------|----------------------|
| **O1** | **Baseline - CNN** | 0.9288 | 1.9027 |
| | *Ablation Components:* | | |
| | *w/ wDCL* | $0.9219 \pm 0.0014$ | $1.9187 \pm 0.0015$ |
| | *w/ MltR-KAN* | $0.9254 \pm 0.0069$ | $1.9174 \pm 0.0025$ |
| | **dcMltR-KAN** | | |
| | *Haar* | $0.9817 \pm 0.0017$ | $1.9936 \pm 0.0009$ |
| | *Mexican Hat* | $0.9804 \pm 0.0016$ | $1.9929 \pm 0.0009$ |
| | *Db4* | $0.9772 \pm 0.0014$ | $1.9916 \pm 0.0005$ |
| | *Sym4* | $0.9772 \pm 0.0031$ | $1.9913 \pm 0.0018$ |
| **O2** | **Baseline - CNN** | 0.9155 | 1.8576 |
| | *Ablation Components:* | | |
| | *w/ wDCL* | $0.8887 \pm 0.0015$ | $1.8154 \pm 0.0014$ |
| | *w/ MltR-KAN* | $0.8850 \pm 0.0059$ | $1.9272 \pm 0.0035$ |
| | **dcMltR-KAN** | | |
| | *Haar* | $0.9799 \pm 0.0072$ | $1.9832 \pm 0.0059$ |
| | *Mexican Hat* | $0.9731 \pm 0.0028$ | $1.9776 \pm 0.0023$ |
| | *Db4* | $0.9744 \pm 0.0076$ | $1.9811 \pm 0.0056$ |
| | *Sym4* | $0.9803 \pm 0.0029$ | $1.9847 \pm 0.0021$ |
| **O3** | **Baseline - CNN** | 0.8363 | 1.8175 |
| | *Ablation Components:* | | |
| | *w/ wDCL* | $0.8888 \pm 0.0008$ | $1.9293 \pm 0.0004$ |
| | *w/ MltR-KAN* | $0.8639 \pm 0.0018$ | $1.8830 \pm 0.0012$ |
| | **dcMltR-KAN** | | |
| | *Haar* | $0.9009 \pm 0.0019$ | $1.9377 \pm 0.0010$ |
| | *Mexican Hat* | $0.9005 \pm 0.0007$ | $1.9126 \pm 0.0007$ |
| | *Db4* | $0.9045 \pm 0.0005$ | $1.9399 \pm 0.0003$ |
| | *Sym4* | $0.9010 \pm 0.0019$ | $1.9378 \pm 0.0012$ |

# 6 Proof of Proposition 1

**Proposition 1: $\mathcal{L}_{\textbf{wdcl}}$ with FNE is lower than the loss without FNE:** $\mathbb{E}_{(x,x(\alpha)^-)}\left[\mathcal{L}_{\text{wdcl}}^{\text{FNE}}\right] < \mathbb{E}_{(x,x^-)}\left[\mathcal{L}_{\text{wdcl}}^{\text{no FNE}}\right]$, where $\alpha$ is the the elimination ratio.

**Statement:** Let $\mathcal{L}_{\text{wdcl}}^{\text{FNE}}(x)$ denote the Wasserstein Debiased Contrastive Loss (wDCL) with False Negatives Elimination (FNE), and $\mathcal{L}_{\text{wdcl}}^{\text{no FNE}}(x)$ denote the wDCL without FNE. Then, under the assumption that the set of negative samples after FNE is a proper subset of the original negative samples, and that the removed negatives are those with the highest similarity to the anchor sample $x$, we have:

$$\mathbb{E}_{(x,x^+,x(\alpha)^-)}\left[\mathcal{L}_{\text{wdcl}}^{\text{FNE}}(x)\right] < \mathbb{E}_{(x,x^+,x^-)}\left[\mathcal{L}_{\text{wdcl}}^{\text{no FNE}}(x)\right], \tag{25}$$

where $\alpha$ is the elimination ratio, $x^-$ are negative samples, and $x(\alpha)^-$ are the negative samples after applying FNE.

*Proof.*

Let $x$ be an anchor sample, $x^+$ its positive counterpart, and $\mathcal{N}$ the set of all negative samples.

Define $\mathcal{N}(\alpha) \subset \mathcal{N}$ as the set after FNE, where the top $\alpha$ fraction of negatives most similar to $x$ are removed.

Let $f$ be the encoder mapping samples to normalized embeddings $\mathbf{h} = f(x)$, $\mathbf{h}^+ = f(x^+)$, and $\mathbf{h}^- = f(x^-)$.

The N-pair contrastive loss without FNE is:

$$\mathcal{L}_{\text{N-pair}}^{\text{no FNE}} = -\log\left(\frac{e^{\mathbf{h}^\top \mathbf{h}^+/\tau}}{e^{\mathbf{h}^\top \mathbf{h}^+/\tau} + \sum\limits_{x^- \in \mathcal{N}} e^{\mathbf{h}^\top \mathbf{h}^-/\tau}}\right). \tag{26}$$

With FNE, it becomes:

$$\mathcal{L}_{\text{N-pair}}^{\text{FNE}} = -\log\left(\frac{e^{\mathbf{h}^\top \mathbf{h}^+/\tau}}{e^{\mathbf{h}^\top \mathbf{h}^+/\tau} + \sum\limits_{x^- \in \mathcal{N}(\alpha)} e^{\mathbf{h}^\top \mathbf{h}^-/\tau}}\right). \tag{27}$$

Since $\mathcal{N}(\alpha) \subset \mathcal{N}$ and the most similar negatives are removed, we have:

$$\sum_{x^- \in \mathcal{N}(\alpha)} e^{\mathbf{h}^\top \mathbf{h}^-/\tau} < \sum_{x^- \in \mathcal{N}} e^{\mathbf{h}^\top \mathbf{h}^-/\tau}. \tag{28}$$

This implies:

$$\frac{e^{\mathbf{h}^\top \mathbf{h}^+/\tau}}{e^{\mathbf{h}^\top \mathbf{h}^+/\tau} + \sum\limits_{x^- \in \mathcal{N}(\alpha)} e^{\mathbf{h}^\top \mathbf{h}^-/\tau}} > \frac{e^{\mathbf{h}^\top \mathbf{h}^+/\tau}}{e^{\mathbf{h}^\top \mathbf{h}^+/\tau} + \sum\limits_{x^- \in \mathcal{N}} e^{\mathbf{h}^\top \mathbf{h}^-/\tau}}. \tag{29}$$

Since $-\log(x)$ is a decreasing function, it follows that:

$$\mathcal{L}_{\text{N-pair}}^{\text{FNE}} < \mathcal{L}_{\text{N-pair}}^{\text{no FNE}}. \tag{30}$$

For the Wasserstein loss $\mathcal{L}_{\text{wass}}$, removing negatives closest to $x$ may increase the distance:

$$\mathcal{L}_{\text{wass}}^{\text{FNE}} \geq \mathcal{L}_{\text{wass}}^{\text{no FNE}}. \tag{31}$$

The total loss difference is:

$$\Delta \mathcal{L} = \mathcal{L}_{\text{wdcl}}^{\text{FNE}} - \mathcal{L}_{\text{wdcl}}^{\text{no FNE}} = \lambda \left( \mathcal{L}_{\text{wass}}^{\text{FNE}} - \mathcal{L}_{\text{wass}}^{\text{no FNE}} \right) - \beta \left( \mathcal{L}_{\text{N-pair}}^{\text{FNE}} - \mathcal{L}_{\text{N-pair}}^{\text{no FNE}} \right). \tag{32}$$

Since $\mathcal{L}_{\text{N-pair}}^{\text{FNE}} < \mathcal{L}_{\text{N-pair}}^{\text{no FNE}}$ (from Equation 30), the second term in Equation 32 is negative. The first term is non-negative due to Equation 31.

By choosing $\beta$ sufficiently large relative to $\lambda$, the decrease in N-pair loss outweighs any increase in Wasserstein loss, ensuring $\Delta \mathcal{L} < 0$.

Taking expectations over the data distribution:

$$\mathbb{E}\left[ \mathcal{L}_{\text{wdcl}}^{\text{FNE}} \right] = \mathbb{E}\left[ \mathcal{L}_{\text{wdcl}}^{\text{no FNE}} + \Delta \mathcal{L} \right] < \mathbb{E}\left[ \mathcal{L}_{\text{wdcl}}^{\text{no FNE}} \right], \tag{33}$$

since $\Delta \mathcal{L} < 0$.

Therefore, we have

$$\mathbb{E}_{(x, x^+, x(\alpha)^-)}\left[ \mathcal{L}_{\text{wdcl}}^{\text{FNE}}(x) \right] < \mathbb{E}_{(x, x^+, x^-)}\left[ \mathcal{L}_{\text{wdcl}}^{\text{no FNE}}(x) \right].$$

# 7 Proof of Proposition 2

The expected WDCL loss with Similarity-Based Weighting (SBW) is lower than without SBW:

$$\mathbb{E}_{(x,\mathbf{v}_i)}\left[\mathcal{L}_{\text{wdcl}}^{\text{SBW}}\right] < \mathbb{E}_{(x,x^-)}\left[\mathcal{L}_{\text{wdcl}}^{\text{no SBW}}\right], \tag{34}$$

where $\mathbf{v}_i$ is the aggregated feature vector from the top $k$ most similar samples via SBW.

## Proof

1. **WDCL Loss Function:**

   The Weighted Decoupled Contrastive Loss (WDCL) for a sample $x$ is defined as:

   $$\mathcal{L}_{\text{wdcl}} = -\log\left(\frac{e^{f(x)^\top f(x^+)/\tau}}{e^{f(x)^\top f(x^+)/\tau} + \sum_{x^-} w(x,x^-)\,e^{f(x)^\top f(x^-)/\tau}}\right), \tag{35}$$

   where:

   - $f(x)$ is the feature representation of sample $x$.
   - $x^+$ is a positive sample associated with $x$.
   - $x^-$ are negative samples.
   - $w(x,x^-)$ is the weight assigned to each negative sample.
   - $\tau$ is a temperature parameter.

2. **Effect of SBW:**

   - **With SBW:** Focuses on the top $k$ most similar negatives, aggregating them into $\mathbf{v}_i$ and assigning appropriate weights.
   - **Without SBW:** Considers a larger set of negatives, often with equal weighting.

3. **Comparison of Denominators:**

   - **With SBW:**

     $$D_{\text{SBW}} = e^{f(x)^\top f(x^+)/\tau} + w_{\text{SBW}} \cdot e^{f(x)^\top \mathbf{v}_i/\tau}, \tag{36}$$

     where $w_{\text{SBW}}$ is the aggregated weight for the negative $\mathbf{v}_i$.
   - **Without SBW:**

     $$D_{\text{no SBW}} = e^{f(x)^\top f(x^+)/\tau} + \sum_{x^-} e^{f(x)^\top f(x^-)/\tau}. \tag{37}$$

4. **Key Observation:**

- **Aggregated Negatives:** SBW's aggregation leads to a more informative negative $\mathbf{v}_i$, but the overall denominator $D_{\text{SBW}}$ grows less than $D_{\text{no SBW}}$.

- **Denominator Size:** A smaller denominator in SBW means the fraction inside the logarithm is larger.

5. **Implication on Loss:** Since the negative logarithm function is decreasing, a larger fraction results in a lower loss:

$$\mathcal{L}_{\text{wdcl}}^{\text{SBW}} < \mathcal{L}_{\text{wdcl}}^{\text{no SBW}}. \tag{38}$$

6. **Expectation over Data:**

Taking expectations over the data distribution confirms the inequality:

$$\mathbb{E}_{(x,\mathbf{v}_i)} \left[ \mathcal{L}_{\text{wdcl}}^{\text{SBW}} \right] < \mathbb{E}_{(x,x^-)} \left[ \mathcal{L}_{\text{wdcl}}^{\text{no SBW}} \right]. \tag{39}$$

Thus, by focusing on the most informative negatives and weighting them appropriately, SBW reduces the expected WDCL loss compared to not using SBW.

# 8 Proof of Proposition 3

The total loss in the Wasserstein Debiased Contrastive Learning (wDCL) framework is a sum over all resolution levels:

$$\mathcal{L}_{\text{wdcl}} = \sum_{l=1}^{L} \omega^{(l)} \mathcal{L}^{(l)}, \tag{40}$$

where:

- $\mathcal{L}^{(l)}$ is the loss at resolution level $l$.

- $\omega^{(l)} \geq 0$ are learned weights adjusting the contribution of each level.

**1. Applying SBW at Each Level Reduces Loss:**
From Proposition 2, we know that applying SBW to the feature representations reduces the expected loss at a single resolution level:

$$\mathbb{E}_{(x, \mathbf{v}_i^{(l)})} \left[ \mathcal{L}^{(l), \text{SBW}} \right] < \mathbb{E}_{(x, x^-)} \left[ \mathcal{L}^{(l), \text{no SBW}} \right], \tag{41}$$

where:

- $\mathbf{v}_i^{(l)}$ is the SBW-refined feature vector at level $l$.

- $\mathcal{L}^{(l), \text{SBW}}$ is the loss at level $l$ with SBW.

- $\mathcal{L}^{(l), \text{no SBW}}$ is the loss at level $l$ without SBW.

**2. Summing Over All Levels:**
Since the inequality holds at each level $l$, we can multiply both sides by the non-negative weights $\omega^{(l)}$ and sum over all levels:

$$\sum_{l=1}^{L} \omega^{(l)} \mathbb{E}_{(x, \mathbf{v}_i^{(l)})} \left[ \mathcal{L}^{(l), \text{SBW}} \right] < \sum_{l=1}^{L} \omega^{(l)} \mathbb{E}_{(x, x^-)} \left[ \mathcal{L}^{(l), \text{no SBW}} \right]. \tag{42}$$

**3. Expressing the Overall Expected Loss:**
The left side represents the overall expected loss with SBW applied:

$$\mathbb{E}_{x_i \sim p_{\text{data}}} \left[ \mathcal{L}_{\text{wdcl}}^{\text{SBW}} \right] = \sum_{l=1}^{L} \omega^{(l)} \mathbb{E}_{(x, \mathbf{v}_i^{(l)})} \left[ \mathcal{L}^{(l), \text{SBW}} \right]. \tag{43}$$

Similarly, the right side is the overall expected loss without SBW:

$$\mathbb{E}_{x_i \sim p_{\text{data}}} \left[ \mathcal{L}_{\text{wdcl}}^{\text{no SBW}} \right] = \sum_{l=1}^{L} \omega^{(l)} \mathbb{E}_{(x, x^-)} \left[ \mathcal{L}^{(l), \text{no SBW}} \right]. \tag{44}$$

As such, combining the above, we have:

$$\mathbb{E}_{x_i \sim p_{\text{data}}} \left[ \mathcal{L}_{\text{wdcl}}^{\text{SBW}} \right] < \mathbb{E}_{x_i \sim p_{\text{data}}} \left[ \mathcal{L}_{\text{wdcl}}^{\text{no SBW}} \right]. \tag{45}$$

This inequality demonstrates that applying SBW before MltR-KAN across all resolution levels reduces the overall expected wDCL loss compared to not applying SBW.

# 9 Visualization of the explainability enhancement process in MltR-KAN

The CNN encoder initially extracts high-level features from the normalized SNR data, which are then decomposed by the Haar wavelet into approximation (cA) and detail coefficients (cD1, cD2), which capture the global and local data behaviors of the SNR feature after CNN. This provides a multi-resolution view of the learned representation, enhancing the transparency and interpretability of the feature extraction process.

The combined use of a CNN encoder followed by Haar wavelet transformation helps us clearly see what features are being learned from the SNR data. The CNN extracts high-level features, while the Haar wavelet further breaks down these features into explainable components, covering both broad trends and finer details. This multi-stage process makes the learned representation more transparent and easier to understand, enhancing explainability.

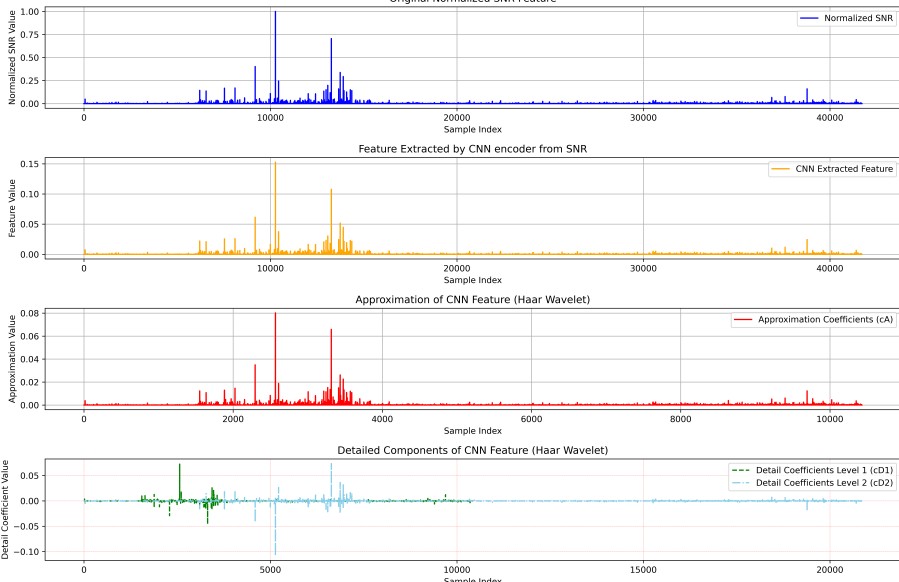

Figure S1: Visualization of the explainability enhancement process in MltR-KAN for the SNR feature from gravitational wave O1 data. The original normalized SNR data is processed by a CNN encoder to extract high-level features. The learned CNN feature is subsequently decomposed using Haar wavelet transformation, resulting in both approximation (cA) and detail coefficients (cD1, cD2), which provide a multi-resolution view of the learned representation, enhancing transparency and interpretability of the feature extraction process

# 10  Impact of False Negative Elimination (FNE) on Hierarchical Loss During Training under MltR-KAN

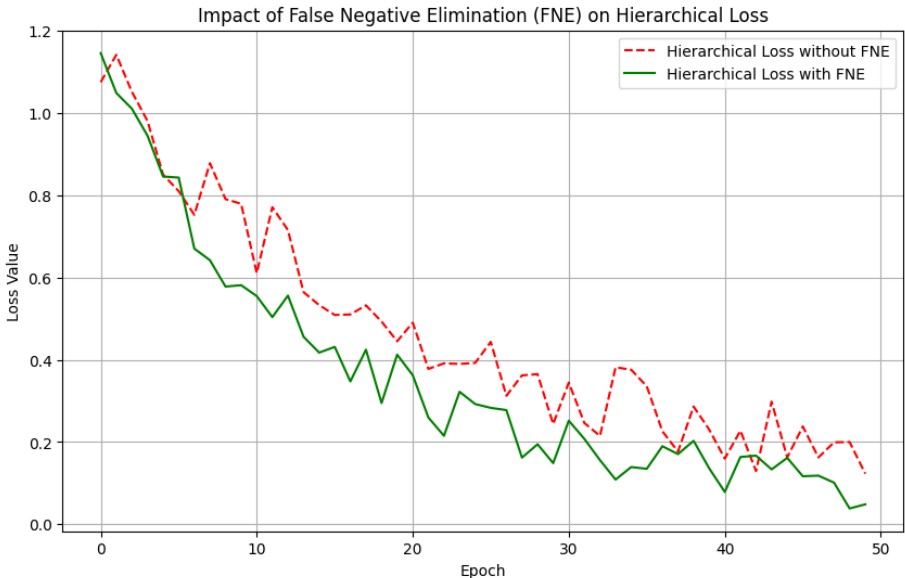

Figure S2: Simulated impact of False Negative Elimination (FNE) on Hierarchical Loss During Training. This figure compares the hierarchical loss values for models trained with and without the False Negative Elimination (FNE) process over 50 epochs. The green line represents the model incorporating FNE, while the red dashed line shows the model without FNE. The model with FNE exhibits a consistently lower loss, indicating that FNE helps to effectively minimize false negatives, leading to enhanced learning and improved convergence during training

# 11 Impact of Similarity-Based Weighting (SBW) on Hierarchical Loss During Training under MltR-KAN

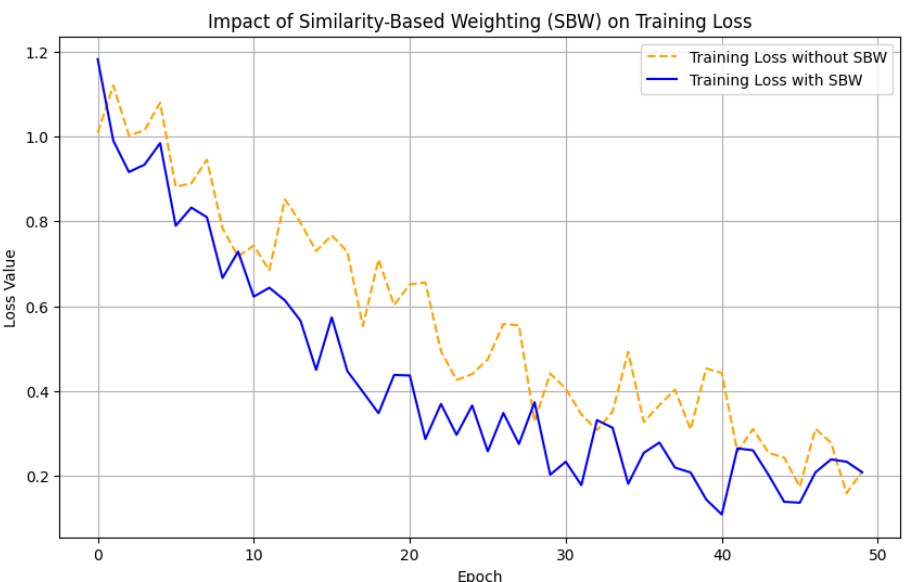

Figure S3:Simulated impact of Similarity-Based Weighting (SBW) on Training Loss. This figure illustrates the effect of incorporating SBW into a simulated training process. The blue line represents the training loss with SBW, while the orange dashed line shows the training loss without SBW. It is evident that using SBW results in a more rapid decline in training loss, indicating enhanced model convergence and efficiency. The reduced overall loss with SBW suggests better feature representation, ultimately contributing to improved model performance.

# 12 Baseline comparisons

## 12.1 CPC(Contrastive Predictive Coding) Result

Contrastive Predictive Coding (CPC) [7] is an unsupervised learning method to extract robust high-level representations from sequential data such as audio, images, text, and reinforcement learning trajectories. The CPC architecture combines an encoder and an autoregressive model to capture temporal or spatial dependencies, encoding input data into a compact latent space that emphasizes essential features while filtering noise. An autoregressive model then processes these encoded representations to create a context vector, preserving temporal relationships and summarizing the information necessary for future predictions. Using a contrastive loss function, specifically InfoNCE, CPC maximizes the mutual information between the context vector and subsequent data, refining its ability to predict future sequence elements. However, CPC has limitations: it is best suited to sequential data, relying on temporal or spatial coherence; it is sensitive to the quality of negative samples, which are essential for effective contrastive learning; and, while it captures broad contextual information, it may overlook finer details.

Table 2: Performance Metrics for Dataset O1, Dataset O2, and Dataset O3

| Metric | Dataset O1 | Dataset O2 | Dataset O3 |
|---|---|---|---|
| Accuracy | 0.775202 | 0.525758 | 0.510133 |
| Precision | 0.773208 | 0.499888 | 0.455736 |
| Recall | 0.775202 | 0.525758 | 0.510133 |
| F1 Score | 0.771829 | 0.475751 | 0.453775 |
| D-index | 1.877014 | 1.696813 | 1.638236 |

## 12.2 TS-TCC(Time-Series Representation Learning via Temporal and Contextual Contrasting) Result

Time-Series Representation Learning via Temporal and Contextual Contrasting (TS-TCC) [3] is an unsupervised framework designed to extract powerful representations from time-series data, which makes it especially effective in scenarios with limited labeled data. By generating two augmented views of the input, one with weaker augmentations and the other with stronger augmentations, TSC learns temporal dependencies by predicting future segments of one view using the context of the other. This cross-view prediction strengthens the model's ability to handle variations from augmentation and capture essential patterns. The contextual contrasting module of TS-TCC further enhances learning by maximizing similarity between contexts of the same sample and minimizing similarity with others, promoting discriminative and generalizable representations. However, TS-TCC demands high computational power due to its use of augmented views and an autoregressive model, and it can be sensitive

to hyperparameters. Additionally, while capturing general temporal patterns effectively, TS-TCC may underperform on tasks that require very fine-grained or specialized features.

Table 3: TS-TCC: Performance Metrics for Dataset O1, Dataset O2, and Dataset O3

| Metric | Dataset O1 | Dataset O2 | Dataset O3 |
|---|---|---|---|
| Accuracy | 0.980205 | 0.976989 | 0.843326 |
| Precision | 0.980719 | 0.977677 | 0.840732 |
| Recall | 0.980205 | 0.976989 | 0.843326 |
| F1 Score | 0.980179 | 0.977055 | 0.839187 |
| D-index | 1.967096 | 1.984084 | 1.819134 |

## 12.3 SimCLR (Simple Contrastive Learning of Representations) Result

SimCLR [1] is a self-supervised framework for learning visual representations, reducing contrastive learning by removing complex architectures and memory banks in favor of large batch sizes and strong enhancements. Train by maximizing agreement between two augmented views of the same image, generated through a data augmentation module that applies transformations such as cropping and color distortion. These views, forming a positive pair, pass through an encoder and projection head to a latent space where contrastive loss aligns similar images. This approach allows SimCLR to achieve performance close to fully supervised models on datasets such as ImageNet. However, SimCLR requires large batch sizes, making it computationally demanding, and its performance heavily depends on carefully chosen augmentations. While strong at capturing general visual features, SimCLR may miss fine details that other, more task-specific methods can capture.

Table 4: SimCLR: Performance Metrics for Dataset O1, Dataset O2, and Dataset O3

| Metric | Dataset O1 | Dataset O2 | Dataset O3 |
|---|---|---|---|
| Accuracy | 0.969796 | 0.966020 | 0.828186 |
| Precision | 0.971033 | 0.968858 | 0.826369 |
| Recall | 0.969796 | 0.966020 | 0.828186 |
| F1 Score | 0.969475 | 0.966591 | 0.822599 |
| D-index | 1.971161 | 1.948001 | 1.826083 |

## 12.4 Fully-supervised deep learning Models

To leverage the time-series structure of the data from all three observing runs (O1, O2, O3), we begin by sorting the dataset chronologically, using earlier data points to train the models and later points to test. Since our dataset is heavily imbalanced, we ensure that both the training and testing sets reflect the same label distribution to maintain a fair performance evaluation across all deep learning models.

For model testing, we split the data, dedicating 80% to training and the remaining 20% to testing. The following machine-learning models were implemented:

- **GAN-DNN Classifier [5]**: This model employs a Generative Adversarial Network (GAN) consisting of a generator and a discriminator to augment the dataset with synthetic samples. The generator network takes random noise as input and produces synthetic data samples, utilizing two dense layers with LeakyReLU activation and batch normalization to stabilize training. The discriminator, structured to classify both real and synthetic samples, has two dense layers with LeakyReLU activation followed by a final dense layer with softmax activation to output labels. The GAN generates 20,000 synthetic samples with three additional labels to balance the original dataset. The final labeled dataset, combining real and synthetic samples, is used for classification training with categorical cross-entropy as the loss function.

- **CNN**: This Convolutional Neural Network (CNN) is designed for sequential data classification. It begins with an input layer that preserves the original shape of the sequence. Two 1D convolutional layers with 64 filters and a kernel size of 3 apply ReLU activation while maintaining the sequence length. The output is flattened and then passed through two dense layers with 64 neurons and ReLU activations, which identify complex patterns. Finally, a softmax output layer, with neurons equal to the target classes, provides class probabilities for classification.

- **Gated Recurrent Unit (GRU) [2]**: This GRU model consists of three layers with 128, 256, and 128 neurons, respectively. Each GRU layer is followed by a dropout layer with rates of 0.1, 0.2, and 0.3. The GRU cells include an update gate and a reset gate, both with sigmoid activation. The update gate controls the balance between the previous hidden state and the current node's hidden state, while the reset gate controls the degree of forgetting of the previous hidden state in calculating the new candidate state. The model ends with a dense output layer that uses softmax activation for class probability output, optimized with categorical cross-entropy.

- **Residual Networks (ResNet) [6]**: A ResNet-50 model is implemented, starting with an initial convolutional layer (64 filters, stride of 2) followed

by batch normalization, ReLU activation, and a max-pooling layer (pool size of 3, stride of 2). The main architecture includes four stages of bottleneck blocks with configurations [3, 4, 6, 3]. Each bottleneck block reduces dimensions, applies a convolution, and then restores dimensions with shortcut connections between the input and output of each block. Batch normalization and ReLU activation are applied throughout. Downsampling occurs at the start of each new stage by adjusting the stride. The model concludes with global average pooling and a dense output layer with softmax activation to produce class probabilities. Categorical cross-entropy is used as the loss function for multiclass classification.

- **Transformer** [8]: This model utilizes a Transformer architecture with a multi-head attention mechanism, configured with 32 heads alongside feed-forward layers. Each Transformer block includes a multi-head attention layer and a feed-forward neural network consisting of dense layers with ReLU activation. Layer normalization is applied both before and after the feed-forward network, while dropout layers are included after the attention and feed-forward layers for regularization. After attention and feed-forward processing, the output is flattened and passed through dense layers for final classification.

Each model was trained for 100 epochs, experimenting with different learning rates (1e-3, 1e-4, 1e-5) and batch sizes (64, 128, 256, 512). The optimal model configuration was selected based on the highest accuracy and D-index, ensuring it did not overfit the training data.

# 13 Silhouette analysis of O1, O2, and O3 data before and after dcMltR-KAN

Table 5: Silhouette analysis under UMAP

| Data | n_neighbors (UMAP) | Silhouette Score (K-Mean clustering) |
|---|---|---|
| Original O1 data | 5 | 0.1847 |
| | 10 | 0.2490 |
| | 15 | 0.1991 |
| | 20 | 0.3094 |
| | 30 | 0.2457 |
| | 50 | 0.2787 |
| O1 data after dcMltR-KAN | 5 | 0.3958 |
| | 10 | 0.5028 |
| | 15 | 0.5307 |
| | 20 | 0.5319 |
| | 30 | 0.5219 |
| | 50 | 0.5401 |
| Original O2 data | 50 | 0.2293 |
| | 60 | 0.2323 |
| | 70 | 0.2139 |
| | 80 | 0.2088 |
| | 90 | 0.1966 |
| | 100 | 0.2328 |
| O2 data after dcMltR-KAN | 50 | 0.4754 |
| | 60 | 0.4963 |
| | 70 | 0.5130 |
| | 80 | 0.4748 |
| | 90 | 0.5125 |
| | 100 | 0.5041 |
| Original O3 data | 50 | -0.0807 |
| | 60 | -0.0181 |
| | 70 | -0.0733 |
| | 80 | -0.0583 |
| | 90 | 0.1428 |
| | 100 | 0.1213 |
| O3 data after dcMltR-KAN | 50 | 0.4317 |
| | 60 | 0.4450 |
| | 70 | 0.4291 |
| | 80 | 0.4237 |
| | 90 | 0.4393 |
| | 100 | 0.4391 |

Note: UMAP is applied to original O1/O2/O3 and their corresponding data after dcMltR-KAN before Kmeans

# 14 dcMltR-KAN results on EMODB and ablation study

Table S5: dcMltR-KAN results on EMODB and ablation study

| Method | Top1 Accuracy (mean ± std) | D-Index (mean ± std) |
|---|---|---|
| *Ablation components:* | | |
| *w/o wDCL* | $0.8503 \pm 0.0277$ | $1.9042 \pm 0.0181$ |
| *w/o MltR-KAN* | $0.8379 \pm 0.0103$ | $1.8955 \pm 0.0067$ |
| **dcMltR-KAN** | | |
| *Mexican-hat* | $0.9326 \pm 0.0035$ | $1.9573 \pm 0.0022$ |
| *Sym4* | $0.9186 \pm 0.0055$ | $1.9483 \pm 0.0035$ |
| *Db4* | $0.9180 \pm 0.0036$ | $1.9478 \pm 0.0023$ |
| *Haar* | $0.8866 \pm 0.0061$ | $1.9278 \pm 0.0040$ |

# 15 Preprocessing and Feature Extraction for EMODB data

The EMODB dataset consists of raw mono audio files, each sampled at 16,000 Hz and approximately two seconds in duration. The audio files were first blocked into small chunks of audio signals, i.e., windowing, where each window has a length of 1024 samples (block size) and is spaced by hop of 512 samples (hop size). For each windowed segment, we extracted features such as Mel Frequency Cepstral Coefficients (MFCC) (first 14 coefficients), spectral centroid, spectral bandwidth, spectral contrast, spectral rolloff, Zero-Crossing Rate (ZCR), Root Mean Square Energy (RMS), and fundamental frequency (F0). Table 1 lists the dimensions of each feature. After feature extraction for each window, we computed two statistics, mean and standard deviation, to represent the overall characteristics of the audio file by aggregating all the instantaneous features. Figure 1 illustrates the preprocessing and feature extraction process.

**Table 1. Audio Dataset Features**

| Features | Feature Dim. for Each Windowed Segment | Aggregated Feature Dim. for Each File |
|---|---|---|
| MFCC | 14 | 28 |
| Spectral Centroid | 1 | 2 |
| Spectral Bandwidth | 1 | 2 |
| Spectral Contrast | 7 | 14 |
| Spectral Rolloff | 1 | 2 |
| Zero-Crossing Rate | 1 | 2 |
| RMS Energy | 1 | 2 |
| $F_0$ | 1 | 2 |

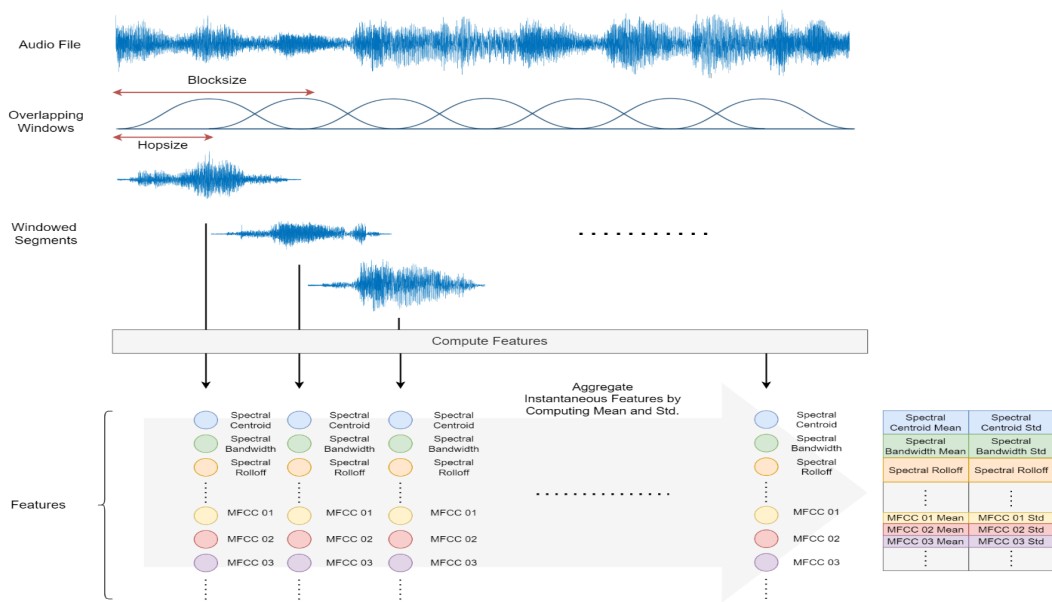

**Figure S1. Feature Extraction of EMODB data.** Each audio file was divided into smaller segments. We then computed the features for each segment as detailed in Table 1. After all features are extracted for each window, we aggregated all these instantaneous features by computing mean and standard deviation to represent the audio file.

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
