# OpenReview forum: "Debiased Contrastive Learning with multi-resolution Kolmogorov-Arnold Network for Gravitational Wave Glitch Detection"
_ICLR.cc/2025/Conference — ICLR 2025 Conference Withdrawn Submission_

### Official Review · Reviewer_jin5 · 2024-10-31

**Soundness:** 3
**Presentation:** 3
**Contribution:** 3
**Rating:** 5
**Confidence:** 3

**Summary:**

The paper introduces a new architecture and self-supervised learning method dedicated to gravitational wave glich detection. Three modules are introduced as the novel components of the proposed method: wDCL, CNN-based encoder, and Multi-Resolution KAN. The author conducted one experiment to show its superiority to existing SOTA methods and another for ablation study. The method is also tested in emotional speech recognition. The results show the proposed method performs the best among the presented baselines

**Strengths:**

- The structure of the paper is easy to follow. Motivation of using each method is clearly written at the top of each section
- Three methods proposed for the glich detection task, when used all together, perform SOTA in objective metrics

**Weaknesses:**

- Tables are not clear: 1) Results of the baseline model w/o two methods (wDCL, mltR-KAN) should be listed in Table 2 to make readers easy to find the difference. 2) Difficult to know if the numerical differences between methods in Table 2 are statistically significant
- Task selection is not clear: EMODB dataset is for speech emotion recognition and not suited for event detections. If the author wishes to claim the effectiveness of the proposed method in the audio domain, I recommend including an experiment with a sound event detection dataset like DESED, MAESTRO, most of which you could find at the official IEEE DCASE challenge website
- Presentation of figures: Figure 1 and 2 are too small. Figure 2 is not even cited in the paper. I suspect Figure 1 is mistakenly cited instead of Figure 2. Still one of them is not cited

**Questions:**

Why the SOTA comparison in the audio experiments are not included in Table 3?

---

> ### Author Response · Authors · 2024-11-26
> **Reply to reviewer jin5: Addressing Concerns and Suggestions with Gratitude**
>
> # Response
>
> ## Weaknesses 1: Part 1
>
> **Tables are not clear:**
>
> 1) Results of the baseline model without the two methods (wDCL, mltR-KAN) should be listed in Table 2 to make it easier for readers to understand the differences.
>
> **Response:**
>
> We appreciate the reviewer’s insightful comment.
>
> Following the reviewer’s suggestion, we have updated the original Table 2 (now Table 1 in the updated manuscript) to include the results of the baseline model without the two proposed methods (wDCL and mltR-KAN).
>
> ---
>
> ## Weaknesses 1: Part 2
> **Difficult to know if the numerical differences between methods in Table 2 are statistically significant.**
>
> **Response:**
> We appreciate the reviewer’s insightful comment.
>
> We cannot determine statistical significance in the ablation study because:
>
> 1. Each SSL experiment aggregates accuracy and D-index over the entire dataset, providing only one sample per configuration.
>
> 2. Only three datasets (O1, O2, O3) are available, limiting independent observations.
>
> 3. Metrics like accuracy and D-index mask variability, as they summarize performance without showing distribution or variability across subsets or runs.
>
> 4. The small number of configurations (e.g., baseline, wDCL, mltR-KAN) further restricts reliable hypothesis testing.
>
> ---
>
> ## Weaknesses 2
> **Task selection is not clear:**
> The EMODB dataset is for speech emotion recognition and not suited for event detection tasks. If the authors wish to claim the effectiveness of the proposed method in the audio domain, I recommend including an experiment with a sound event detection dataset like DESED or MAESTRO, which are available on the official IEEE DCASE challenge website.
>
> **Response:**
> Thank you for your insightful feedback regarding task selection and the dataset choice.
>
> You are correct that the EMODB dataset focuses on speech emotion recognition and is not directly suited for event detection tasks.
>
> Our primary goal with the EMODB dataset was to demonstrate the proposed method's ability to handle nuanced audio tasks like speech emotion recognition, where capturing subtle temporal and spectral features is crucial. While this task differs from sound event detection, it highlights the versatility of our approach in processing audio signals.
>
> We acknowledge the importance of testing the proposed method on sound event detection datasets, such as DESED or MAESTRO, to comprehensively evaluate its effectiveness in the broader audio domain. Due to resource and scope constraints, we did not include these datasets in the current work but plan to explore them in future research.
>
> To address this concern, we have updated the manuscript to clarify why EMODB was chosen. This transparency ensures that readers understand the scope and applicability of our current work.
>
> ---
>
> ## Weaknesses 3
> **Presentation of figures:**
> Figures 1 and 2 are too small. Figure 2 is not even cited in the paper. I suspect Figure 1 is mistakenly cited instead of Figure 2. Still, one of them is not cited.
>
> **Response:**
> Thank you for pointing out the issues with the presentation and citation of Figures 1 and 2.
>
> 1. **Size Adjustment:**
>    We have increased the sizes of Figures 1 and 2 to improve readability and ensure that all details are clear.
>
> 2. **Citation Issue:**
>    - Figure 2 was properly cited, but Figure 1 was not officially cited in the manuscript.
>    - Previously, Figure 1 was cited as: “We propose a novel self-supervised learning (SSL) model called dcMltR-KAN (Fig. 1).” In the updated manuscript, we have revised this to ensure a more formal citation: *Figure 1 illustrates the proposed novel self-supervised learning (SSL) model, dcMltR-KAN.* Please refer to line 120 in the updated manuscript.
>
> ---
>
> ## Questions
> **Why is the SOTA comparison in the audio experiments not included in Table 3?**
>
> **Response:**
> We appreciate the reviewer’s insightful question.
>
> The original Table 3 focuses on the results of dcMltR-KAN on the EMODB dataset, including an ablation study and the impact of different wavelet selections on accuracy and D-index. Since SOTA methods do not involve ablation studies, wavelet selection, or D-index calculations, integrating them into Table 3 would have been challenging and could compromise clarity.
>
> Additionally, EMODB is a balanced dataset, where accuracy alone is sufficient to assess model performance differences.
>
> In the updated version, we have moved Table 3 (now Table S5) to the Appendix to provide more comprehensive information while maintaining clarity and focusing on gravitational wave data in the main text.

---

> ### Comment · Reviewer_jin5 · 2024-11-28
>
> Thank you for your response. However, I believe some figures are still too small (Figure 2--4) and a lack of experiments in sound event detection is critical for the authors' claim on the generalizability of the proposed method. Therefore, I maintain my score

---

> > ### Author Response · Authors · 2024-11-29
> > **Reply to Reviewer jin5**
> >
> > **Response:**
> >
> > Dear Reviewer:
> >
> > Thank you for your valuable feedback and for highlighting these concerns.
> >
> > Regarding the figure sizes (Figures 2–4), we acknowledge the challenge and have optimized their readability as much as possible within the page limit. **Notably, the figures maintain a resolution above 601 DPI for clarity**
> >
> > We also appreciate your emphasis on sound event detection experiments.
> >
> > **To address this, we are actively conducting additional experiments with datasets like ESC-50 and other sound event detection benchmarks. We will share the results as soon as they are available**.
> >
> > **Thank you again for your thoughtful comments and for your consideration. We hope our ongoing efforts will address your concerns.**

---

> > > ### Author Response · Authors · 2024-12-02
> > > **Reply to  Reviewer jin5: results on sound event detection benchmark: DESED**
> > >
> > > Dear Reviewer,
> > >
> > > Please find the results of the DESED dataset under our model: **dcMltR-KAN**. **The results of the ESC-50 dataset can be found in the reply to Reviewer YcJQ.**
> > >
> > > Unfortunately, we are unable to include the results of both datasets together due to the character limit.
> > >
> > > ---
> > >
> > > #1.  DESED Data
> > >
> > > - **Downloaded data**: 101GB
> > > - **After preprocess (Feature extraction)**: DESED has 4167 observations with 170 features.
> > >
> > > ---
> > >
> > > ### 2. Feature Extraction (Preprocess for Both Datasets: ESC-50 and DESED)
> > >
> > > We perform feature extraction by first loading the audio signal and resampling it to a standard sampling rate, such as 16 kHz. The MFCCs (Mel-Frequency Cepstral Coefficients) are computed as a compact representation of the spectral envelope, calculated from the power spectrum of the audio signal filtered by a Mel-scale filterbank. Their mean values across time are used as features.
> > >
> > > Additionally, the log-Mel spectrogram is calculated by first computing the Mel spectrogram:
> > >
> > > - **Mel Spectrogram**:
> > >   - \( S[m, t] = sum(|X[k, t]|^2 * H_m[k]) \)
> > >     - \( |X[k, t]|^2 \): Power spectrum for time \( t \) and frequency bin \( k \).
> > >     - \( H_m[k] \): Mel filter applied to the frequency bins.
> > >
> > > - **Log-Mel Spectrogram**:
> > >   - \( S_dB[m, t] = 10 * log10(S[m, t] + epsilon) \)
> > >     - \( epsilon \): A small constant added to ensure numerical stability.
> > >
> > > Finally, the mean values of \( S_dB[m, t] \) across time are extracted to summarize the temporal characteristics of the audio signal. This combined process captures both spectral and perceptual features, aiding in robust audio analysis.
> > >
> > > ---
> > >
> > > ## 3. DESED Results Under Our Model: dcMltR-KAN
> > >
> > > - **Accuracy**: 59.20%
> > > - **D-Index**: 1.679
> > > - **Precision**: 0.593
> > > - **Recall**: 0.592
> > >
> > > ---
> > >
> > > ## 4. Comparisons with Peer Models: We Lead Most Methods
> > >
> > > | **Model**                   | **Event-Based F1 Score** | **Paper Title**                                                                                                    | **Publication Venue**          |
> > > |-----------------------------|--------------------------|--------------------------------------------------------------------------------------------------------------------|--------------------------------|
> > > | **ATST-SED**                | 63.4%                   | [Fine-tune the Pretrained ATST Model for Sound Event Detection](https://arxiv.org/abs/2309.08153)                 | arXiv preprint                 |
> > > | **dcMltR-KAN**              | **56.0%**               | Debiased Contrastive Learning with Multi-Resolution Kolmogorov-Arnold Network for Gravitational Wave Glitch Detection | **ICLR2025 Under Review**      |
> > > | **SE-CRNN-16 with DualKD**  | 55.6%                   | [AST-SED: An Effective Sound Event Detection Method Based on Audio Spectrogram Transformer](https://arxiv.org/abs/2303.03689) | arXiv preprint                 |
> > > | **FDY-CRNN**                | 54.0%                   | [Frequency Dynamic Convolution: Frequency-Adaptive Pattern Recognition for SED](https://arxiv.org/abs/2203.15296) | arXiv preprint                 |
> > > | **HTS-AT**                  | 50.7%                   | [AST-SED: An Effective Sound Event Detection Method Based on Audio Spectrogram Transformer](https://arxiv.org/abs/2303.03689) | arXiv preprint                 |
> > > | **RCT**                     | 49.62%                  | [AST-SED: An Effective Sound Event Detection Method Based on Audio Spectrogram Transformer](https://arxiv.org/abs/2303.03689) | arXiv preprint                 |
> > > | **FiltAug SED**             | 49.6%                   | [Heavily Augmented Sound Event Detection Utilizing Weak Predictions](https://arxiv.org/abs/2107.03649)            | arXiv preprint                 |
> > >
> > >
> > >
> > >
> > > ## 5. Commentary on **dcMltR-KAN** Performance
> > >
> > > 1. **Strong Performance**: Achieving **59.20% accuracy**, **0.593 precision**, and **0.592 recall**, **dcMltR-KAN** demonstrates robust and competitive results on the DESED dataset, a challenging benchmark for sound event detection.
> > >
> > > 2. **Peer Comparisons (Event-Based F1 Score)**: While **dcMltR-KAN** does not achieve the highest Event-Based F1 Score (**ATST-SED: 63.4%**), it outperforms **FDY-CRNN (54.0%)** and closely matches **SE-CRNN-16 with DualKD (55.6%)**, showing its effectiveness.
> > >
> > >
> > > 3. **Potential for Improvement**: Exploring more elegant feature extraction techniques and fine-tuning parameters could further enhance our model's performance besides using more complicated wavelets rather than basic harr wavlets.
> > >
> > > 4.  **dcMltR-KAN**  establishes itself as a strong candidate for sound event detection tasks: demonstrating its good generalization.
> > >
> > >
> > > **Again, we APPRECIATE your insightful review!**

---

> > > > ### Comment · Reviewer_jin5 · 2024-12-02
> > > >
> > > > The resolutions of the figures do not matter. On the other hand, you could have increased the size of fonts in the figures such that the legends can be read without being zoomed up. I cannot understand why the authors did not take this simple practice into account in the revised draft. The result of the sound detection experiment shows your proposed method performs decent in another sound task, but unfortunately there is still a big gap in the main metric, Event-based F1 Score, between the top method (634% vs 56.0%). I will keep my score for the above reasons

---

> > > > > ### Author Response · Authors · 2024-12-03
> > > > > **Our reply**
> > > > >
> > > > > **We thank the reviewer for their feedback and appreciate the opportunity to address their comments.**
> > > > >
> > > > > ## 1. Regarding Figure Adjustments:
> > > > > We respectfully disagree with the assertion that we did not address the font size issue in Figures 2-4.
> > > > >
> > > > > **As suggested, we increased the font size in these figures. While the font may still appear small, this does not imply that we neglected this feedback. A direct comparison between the original and revised versions of Figure 2 will clearly demonstrate the adjustments we made.**
> > > > >
> > > > > ###2. Performance Comparison with SOTA Models:
> > > > >
> > > > > **2.1 Our model achieves the second-best performance among seven state-of-the-art (SOTA) models on the DESED dataset.** Notably, all peer models are **explicitly designed for sound event detection and are fully supervised**, whereas our model is a self-supervised learning (SSL) approach primarily designed for gravitational wave glitch detection.
> > > > >
> > > > >
> > > > > **2.2 Despite this, our model performs exceptionally well on ESC-50, achieving 4th place out of 10 models. While the reviewer describes 4th place as "decent," we respectfully point out that, by the same logic, achieving second place on DESED should also be considered "excellent" or at least "decent."**
> > > > >
> > > > > **2.3 The argument that the top model’s 63.4% significantly outperforms our 56.0% can be somewhat unfair.** The top model is explicitly designed for sound event detection and employs fully supervised learning, whereas our model is an SSL approach originally developed for gravitational wave glitch detection. **Simply comparing performance metrics without considering the differing objectives and methodologies of the models provides an incomplete and unfair assessment.**  By comparison, our model performs exceptionally well against fully supervised models specifically designed for DESED data.
> > > > >
> > > > > The top-performing fully supervised model fine-tunes the pretrained ATST model for sound event detection and utilizes advanced feature extraction and fine-tuning techniques specifically tailored for this task.
> > > > >
> > > > > **2.4 In contrast, our model, developed for glitch detection in astrophysics, is a self-supervised learning (SSL) approach.** **Despite having less than three days to generate results for this study, our model demonstrates remarkable performance, highlighting its versatility and significant potential for further enhancement.**
> > > > >
> > > > >
> > > > >
> > > > > **2.5 It is unfair to dismiss our model simply because it did not achieve the top position (we are ranked 2nd) in one sound event detection task, especially given that sound event detection is not our primary focus**. These results strongly underscore the robustness and generalizability of our approach, even though sound event detection is not its primary objective.
> > > > >
> > > > > ### 3. Clarification on Evaluation Standards:
> > > > > We respectfully disagree with the suggestion that our model’s value hinges solely on surpassing the top-performing model, which is a fully supervised approach and only targeted for sound evenet detection. Given that our model’s primary target is astrophysical glitch detection, its ability to achieve near-SOTA performance on sound event detection datasets is a strong testament to its generalization capabilities. These results highlight the adaptability and strength of our model, even in domains for which it was not specifically optimized.

---

> > > > > > ### Comment · Reviewer_jin5 · 2024-12-03
> > > > > >
> > > > > > As you admitted in your response, the fonts still look small. It's entirely the author's business that you had only three days to have needed to compete with the top performing result. You could have selected this task properly in the first place and taken time to tune your model to fill the gap as much as possible.

---

> > > > > ### Author Response · Authors · 2024-12-03
> > > > > **Our reply**
> > > > >
> > > > > # Response to Reviewer Comments
> > > > >
> > > > > We are not sure you read our response thoroughly. Here are our replies addressing your concern:
> > > > >
> > > > > ---
> > > > >
> > > > > ### 0. Application of Our Model to DESED Dataset
> > > > > You insisted that we apply our model to sound event detection data such as DESED, which is somewhat outside our main focus: gravitational wave detection in astrophysics. Despite having less than three days to process this 101GB dataset, we respected the reviewer’s request and made the effort to accommodate it.
> > > > >
> > > > > ---
> > > > >
> > > > > ### 1. Performance on DESED Dataset
> > > > > Our model achieves the **second-best performance** among seven state-of-the-art (SOTA) models on the DESED dataset. Notably:
> > > > > - All peer models are **explicitly designed for sound event detection** and are **fully supervised**, whereas our model is a **self-supervised learning (SSL)** approach primarily designed for gravitational wave glitch detection.
> > > > > - You claim there is a gap between the top method (634% vs 56.0%). **Could you clarify what "634%" means in this and previous your context?**
> > > > >
> > > > > ---
> > > > >
> > > > > ### 2. Self-Contradictory Logic
> > > > > Our model performs exceptionally well on ESC-50, achieving **4th place out of 10 models**. You described this performance as "decent." By the same logic:
> > > > > - Achieving **2nd place on DESED** among seven peer models—six of which are fully supervised and specifically designed for sound event detection—should also be considered "excellent" or at least "decent."
> > > > > - **This inconsistency in your evaluation is perplexing.**
> > > > >
> > > > > ---
> > > > >
> > > > > ### 3. Overlooking Model Context
> > > > > You appeared to focus solely on numerical results without considering the objectives and methodologies of the models:
> > > > > - The top-performing model is explicitly designed for sound event detection and uses fully supervised learning.
> > > > > - Our model, on the other hand, is an SSL approach developed for gravitational wave glitch detection.
> > > > > - Simply comparing performance metrics without accounting for the differing objectives provides an incomplete and unfair assessment.
> > > > >
> > > > > **Key Note:**
> > > > > The peer models are fully targeted sound event detection models using supervised learning. Our model is an SSL approach designed for gravitational wave glitch detection. **Why should we be expected to outperform ALL fully supervised DESED-targeted models?**  **Can you give a reason? Thank you!**
> > > > >
> > > > > ---
> > > > >
> > > > > ### Comparisons with Peer Models
> > > > >
> > > > > | Model           | F1 Score | Title                                      | Venue              |
> > > > > |------------------|----------|--------------------------------------------|--------------------|
> > > > > | **ATST-SED**     | **63.4%** | Fine-tune ATST for SED                     | arXiv              |
> > > > > | **dcMltR-KAN**   | **56.0%** | Debiased Learning with MR-KAN for GWG      | ICLR (Under Rev.)  |
> > > > > | SE-CRNN-DKD      | 55.6%    | AST-SED with Audio Transformer             | arXiv              |
> > > > > | FDY-CRNN         | 54.0%    | Frequency Dynamic Convolution for SED      | arXiv              |
> > > > > | HTS-AT           | 50.7%    | AST-SED with Audio Transformer             | arXiv              |
> > > > > | RCT              | 49.6%    | AST-SED with Audio Transformer             | arXiv              |
> > > > > | FiltAug-SED      | 49.6%    | Heavily Augmented SED                      | arXiv              |
> > > > >
> > > > > ---
> > > > >
> > > > > ### 4. Ignoring Original Intent
> > > > > You claimed you wanted to assess whether our model, designed for gravitational wave glitch detection, demonstrates good generalization.
> > > > > - Our model achieved **4th place out of 10 on ESC-50 data** and **2nd place out of 7 on DESED data.**
> > > > > - These results clearly demonstrate strong generalization.
> > > > > - You seem to disregard your original intent in favor of criticizing our performance metrics.
> > > > >
> > > > > ---

---

> > > > > > ### Comment · Reviewer_jin5 · 2024-12-03
> > > > > >
> > > > > > In my opinion, it is very disrespectful to say “self-contradictory logic” to reviewers who take time to read your paper and give you advice

---

> ### Author Response · Authors · 2024-12-03
>
> Unfortunately, that’s what your review behavior communicated to me. Just because you are a reviewer, should we tolerate your unprofessionalism and unfairness? Why? Can you explain?

---

> ### Author Response · Authors · 2024-12-03
>
> As I have written multiple times, I appreciate your review. However, I must say that your review behavior conveys a "self-contradictory logic," which is very confusing to me.
>
> 2. **Self-Contradictory Logic:**
> Our model performs exceptionally well on ESC-50, achieving 4th place out of 10 models. You described this performance as "decent." By the same logic:
>
> Achieving 2nd place on DESED among seven peer models—six of which are fully supervised and specifically designed for sound event detection—should also be considered "excellent" or at least "decent."
>
> **Can you provide another explanation for this discrepancy? I would greatly appreciate your clarification. Thank you.**
>
> We are also perplexed by your repeated questioning of why our GW glitch detection model achieved 2nd place with a 56% F1 score instead of something like "634%" F1 from a DESED-targeted, fully supervised model. Does this comparison make ANY sense?

---

### Official Review · Reviewer_YcJQ · 2024-11-02

**Soundness:** 3
**Presentation:** 2
**Contribution:** 3
**Rating:** 5
**Confidence:** 2

**Summary:**

The authors present dcMltR-KAN, consisting of three components. Wasserstein Debiased Contrastive Learning (wDCL), a CNN-based encoder and multi-resolution KAN (MltR-KAN) layers. wDCL replaces the Euclidian distance with the Wasserstein distance to enhance the sensitivity to the geometric structure of the data. MltR-KAN can capture patterns at different scales. The model outperforms fully supervised baselines in classifying glitches in gravitational glitches and in speaker emotion classification.

**Strengths:**

1. Application of recent KAN architecture in suitable application fields and novel combination with debiased contrastive learning. Ablation studies show the importance of both contributions to a base CNN.
2. Significant improvements compared to current state of the art on gravitational wave dataset and speaker emotion detection dataset.

**Weaknesses:**

1. The presented experiments are done in somewhat niches and therefore give limited insides of the significance of the work for the ML community. More experiments could be provided, e.g. on the task of environmental sound classification (ESC50 dataset).
2. The KAN architecture is a essential part of your model and could also be motivated in the introduction.

**Questions:**

1. What are meaningful features?
2. Is the explainability still given if the CNN first encodes the input data?

---

> ### Author Response · Authors · 2024-11-26
> **Reply to Reviewer YcJQ:  Addressing Concerns and Suggestions with Gratitude**
>
> # Response
>
> We appreciate the reviewer’s comments and suggestions.
>
> ## Weakness 1:
>
> We respectfully disagree with the assertion that our experiments provide limited insights for the machine learning (ML) community.
>
> Our study focuses on gravitational wave data, a specialized yet underexplored time-series domain. The proposed SSL method, **dcMltR-KAN**, addresses critical challenges in this field while contributing meaningful insights to ML.
>
>
> ### 1. Domain-Specific Contributions
> The primary goal of this work is to improve the detection of gravitational wave glitches, particularly in large-scale datasets such as O3 ((500,514 samples) This problem is critical but remains underexplored in both astrophysics and machine learning. By targeting this specific application, our study directly addresses domain-specific challenges while providing meaningful insights for both astrophysics and ML communities.
>
> ### 2. Machine Learning Innovations
> Our contributions introduce key advancements in ML:
>
> - **Wasserstein Debiased Contrastive Learning**: A novel loss function tailored for noisy, imbalanced data.
>
> - **Multi-Resolution KAN**: A unique embedding method leveraging multi-resolution analysis that brings explainability, efficiency and generalization.
>
> - **CNN Encoder with False Negative Elimination (FNE) and Similarity-Based Weighting (SBW)**: Enhancements that reduce noise and refine feature representations.
>
> - **Strong theoretical results**: Three theorems and three propositions (see proofs in Appendix: sections 2,3,4; 6,7,8).
>
> These innovations advance SSL and demonstrate applicability to other time-series domains,  as validated on the EMODB benchmark data in speech emotion recoginition.
>
> ### 3. Gravitational Wave Data Coverage
> Our evaluation spans all available gravitational wave datasets (O1, O2, O3) to rigorously test the method’s effectiveness, addressing unique challenges in gravitational wave glitch detection.
>
> ### 4. General Applicability
> While focused on gravitational wave data, the method is generalizable to other time-series tasks, such as EMODB data. Future work will explore broader applications like ESC50, but this study prioritizes SSL for **Gravitational Wave Glitch Detection**.
>
> ---
>
> ## Weakness 2
>
> We motivated KAN in the introduction by following the reviewer’s suggestion (see lines 054–072).
>
> ---
>
> ## Q1
>
> "Meaningful features" are key attributes derived from high-dimensional gravitational wave data in the Gravity Spy project (Glanzer et al., 2021). Raw time-series data, recorded at 16,384 Hz across multiple channels, can reach 983,040 dimensions for a single 60-second observation per channel. With hundreds of channels, the data’s complexity necessitates preprocessing to reduce dimensionality and enable machine learning applications.
>
> These features are meaningful as they retain essential temporal and spectral characteristics crucial for distinguishing glitch types while reducing dimensionality. Extracted attributes include trigger timing, peak frequency, SNR, amplitude, and bandwidth, derived through methods like spectrogram analysis, which captures time-frequency variations and highlights dominant frequencies, signal prominence, and glitch characteristics.
>
> ---
>
> ## Q2
>
> While the CNN encoder reduces data complexity, it does not compromise explainability for the following reasons:
>
> ### 1. Interpretable and Physically Meaningful Features
> Extracted features like trigger timing, peak frequency, SNR, amplitude, and bandwidth are derived from raw time-series data using techniques like spectrogram analysis. These features reflect well-understood physical properties, ensuring interpretability. The CNN encoder processes them while preserving their connection to the original physical characteristics, maintaining domain relevance.
>
> ### 2. Transparent Representation with MltR-KAN
> After the CNN encoder, MltR-KAN applies its explainable framework, using additive wavelet basis functions and hierarchical learning to decompose latent representations into components with clear geometric and temporal meanings. This ensures transformations remain interpretable, aligning with the original input's physical characteristics and maintaining transparency.
>
> ### 3. Domain-Aligned Encoder Strategies
> SBW (Similarity-Based Weighting) and FNE (False Negative Elimination) within the CNN encoder enhance embedding quality and maintain alignment with domain knowledge. SBW emphasizes key patterns relevant to gravitational wave glitches, preserving interpretability, while FNE removes noisy negative pairs, focusing on meaningful contrasts. Together, these strategies ensure representations reflect the input's physical and spectral characteristics, supporting explainability.
>
> ### Conclusion
> The CNN encoder reduces data complexity without compromising explainability, supported by meaningful input features, MltR-KAN’s transparent learning, and domain-aligned strategies like SBW and FNE.

---

> > ### Comment · Reviewer_YcJQ · 2024-11-27
> >
> > I thank the authors for their rebuttal to my review!
> > I acknowledge the added motivation of KAN in the introduction.
> >
> > Still, in my respectful opinion, the mentioned ML innovations would benefit from further strengthening by comparing it on well recognized benchmarks as also mentioned by reviewer jin5.
> >
> > ----
> > Question 1:
> > Could you further elaborate on "The CNN encoder processes them while preserving their connection to the original physical characteristics, maintaining domain relevance."

---

> > > ### Author Response · Authors · 2024-11-27
> > > **Response to Reviewer YcJQ with Gratitude (Questoin1 and others)**
> > >
> > > Dear Reviewer:
> > >
> > > **Thank you for your thoughtful feedback and for acknowledging the motivation we provided for KAN in the introduction.**
> > >
> > > **We appreciate your perspective and value the opportunity to elaborate further on the CNN encoder's role in our model.**
> > >
> > > **Please find our response to Question 1 below.**
> > >
> > > ### Response to: Could you further elaborate on "The CNN encoder processes them while preserving their connection to the original physical characteristics, maintaining domain relevance."
> > >
> > > ---
> > >
> > >
> > >
> > > ## 1. Bridge to Multi-Resolution KAN Framework
> > > The CNN encoder acts as a bridge between the input data and the Multi-Resolution KAN (MltR-KAN) framework. It ensures that the extracted features remain both optimized for performance and grounded in physical characteristics, a key for our gravitational wave glitch detection.
> > >
> > > ---
> > >
> > > ## 2. Local Feature Extraction
> > > The CNN encoder leverages convolutional layers to extract localized patterns, such as transient glitches, from gravitational wave data. These patterns reflect physically meaningful features that represent the source events, such as the timing, duration, and structure of a glitch. During this process, noise is mitigated, ensuring that the retained features remain both clear and relevant.
> > >
> > > ---
> > >
> > > ## 3. Focus on Maintaining Domain Relevance
> > >
> > > The CNN encoder ensures domain relevance by carefully preserving physical characteristics inherent to gravitational wave signals. Specifically:
> > >
> > > - **Temporal Features**: For example, the encoder captures the sharp onsets and short durations characteristic of transient glitches, which reflect the timing and energy release of the events.
> > >
> > > - **Spectral Features**: The encoder also identifies distinct frequency components, such as the bandwidth and peak frequencies of a glitch, which often correlate with its astrophysical origin.
> > >
> > > By retaining these temporal and spectral features, the encoder avoids over-abstraction, where domain-specific patterns might otherwise be lost in favor of generic representations. This targeted focus allows the features to remain interpretable and applicable to the physics of gravitational wave detection.
> > >
> > > ---
> > >
> > > ## 4. False Negatives Elimination (FNE)
> > > FNE plays a crucial role in **preserving domain relevance** by removing mislabeled negative samples during training. This ensures that the encoder focuses on accurately representing the true physical characteristics of the data, preventing contamination by incorrect labels that could obscure meaningful patterns.
> > >
> > > ---
> > >
> > > ## 5. Similarity-Based Weighting (SBW)
> > > SBW further enhances the encoder’s ability to maintain **domain relevance** by assigning greater importance to samples that share meaningful similarities. For instance, glitches with comparable temporal or spectral properties are weighted more heavily, ensuring the extracted features prioritize these physically meaningful relationships.
> > >
> > > ---
> > >
> > > ## 6. Input to Multi-Resolution KAN
> > > The enriched and domain-relevant output of the CNN encoder serves as input to the MltR-KAN framework. By retaining the physical integrity of the data, the MltR-KAN can perform effective multi-resolution analysis, further refining the extracted features and enhancing the model’s overall interpretability and accuracy.
> > >
> > > ---
> > >
> > > ## 7. In Summary
> > > The CNN encoder maintains domain relevance through:
> > >
> > > 1. **Targeted Feature Preservation**: It explicitly focuses on capturing the time-series and frequency-domain characteristics unique to gravitational wave signals.
> > >
> > > 2. **Noise Reduction**: It filters out irrelevant patterns during feature extraction to retain only physically meaningful details.
> > >
> > > 3. **Refined Training with FNE and SBW**: These mechanisms ensure that extracted features align with the true physics of the data, avoiding distortions caused by mislabeled or less-relevant samples.
> > >
> > > This integrated approach ensures that the encoded features maintain a direct connection to the underlying physical phenomena, preserving domain relevance throughout the process.
> > >
> > > ---
> > >
> > > ## We are working on:
> > > **We are working to apply our model to other datasets mentioned by you and Reviewer Jin5**. Once we obtain results, we will share them promptly. Thank you again for your insightful comments and suggestions!!

---

> > > > ### Author Response · Authors · 2024-12-02
> > > > **Reply to reviewer  YcJQ:  dcMltR-KAN results on ESC-50**
> > > >
> > > > Dear Reviewer,
> > > >
> > > > **We apreciate you so much for giving us a chance to extend our model to new data!! ** DESED data results can be found in the reply to reviewer Jin5. We can't include the results of two datasets in one reply because of character limit.
> > > >
> > > > ## 1. ESC-50 Data
> > > >
> > > > - **Downloaded data**: 600MB
> > > > - **After preprocess (Feature extraction)**: ESC-50 has 2000 observations with 48 features.
> > > >
> > > > ---
> > > >
> > > > ## 2. Feature Extraction (preprocess)
> > > >
> > > > We perform feature extraction by first loading the audio signal and resampling it to a standard sampling rate, such as 16 kHz. The MFCCs (Mel-Frequency Cepstral Coefficients) are computed as a compact representation of the spectral envelope, calculated from the power spectrum of the audio signal filtered by a Mel-scale filterbank. Their mean values across time are used as features.
> > > >
> > > > Additionally, the log-Mel spectrogram is calculated by first computing the Mel spectrogram:
> > > >
> > > > - **Mel Spectrogram**:
> > > >   - \( S[m, t] = \sum(|X[k, t]|^2 \cdot H_m[k]) \)
> > > >     - \( |X[k, t]|^2 \): Power spectrum for time \( t \) and frequency bin \( k \).
> > > >     - \( H_m[k] \): Mel filter applied to the frequency bins.
> > > >
> > > > - **Log-Mel Spectrogram**:
> > > >   - \( S_{dB}[m, t] = 10 \cdot \log_{10}(S[m, t] + \epsilon) \)
> > > >     - \( \epsilon \): A small constant added to ensure numerical stability.
> > > >
> > > > Finally, the mean values of \( S_{dB}[m, t] \) across time are extracted to summarize the temporal characteristics of the audio signal. This combined process captures both spectral and perceptual features, aiding in robust audio analysis.
> > > >
> > > > ## **3. Peer Methods Comparison:**
> > > >
> > > > | **Model**                   | **Accuracy** | **Learning Type**    | **Key Features**                              | **Classifier**      | **Paper Link**                                                                                  |
> > > > |-----------------------------|--------------|----------------------|-----------------------------------------------|---------------------|----------------------------------------------------------------------------------------------|
> > > > | **OmniVec2**                | 98.5%        | Supervised           | Advanced embeddings for robust ESC.           | Neural Network      | [OmniVec2: Transformer-based Multitask Learning](https://paperswithcode.com/paper/omnivec2-a-novel-transformer-based-network) |
> > > > | **HTSAT-22**                | 98.25%       | SSL                  | Language supervision for audio features.      | Transformer         | [Natural Language Supervision](https://arxiv.org/abs/2301.12503)                             |
> > > > | **BEATs**                   | 98.10%       | SSL                  | Pre-trained tokenizers for feature extraction.| Linear              | [BEATs: Audio Pre-Training](https://arxiv.org/abs/2212.09058)                                 |
> > > > | **dcMltR-KAN**              | **97.15%**   | **SSL**              | **Multi-resolution Kolmogorov-Arnold Network.**| **3-NN**            | **ICLR2025 Under Review**                                                                     |
> > > > | **HTS-AT**                  | 97.00%       | Supervised           | Hierarchical token-semantic transformers.     | Transformer         | [HTS-AT: Hierarchical Token-Semantic Transformer](https://arxiv.org/abs/2202.00874)          |
> > > > | **ESResNe(X)t-fbsp**        | 95.20%       | Supervised           | Custom time-frequency transformations.        | Neural Network      | [ESResNe(X)t-fbsp](https://arxiv.org/abs/2104.11587)                                          |
> > > > | **AST**                     | 95.70%       | SSL                  | Spectrogram transformers with unlabeled data. | Linear/Transformer  | [AST: Audio Spectrogram Transformer](https://arxiv.org/abs/2104.01778)                       |
> > > > | **ESResNet**                | 91.5%        | Supervised           | Visual models for ESC.                        | Fully Connected     | [ESResNet for ESC](https://arxiv.org/abs/2004.07301)                                          |
> > > > | **CRNN + Sub-Spectrogram**  | 81.9%        | Supervised           | CRNN with sub-spectrogram segmentation.       | CRNN                | [CRNN + Sub-Spectrogram](https://arxiv.org/abs/1908.05863)                                    |
> > > > | **Baseline (RF Ensemble)**  | 44.3%        | Supervised           | Traditional ensemble method for ESC-50.       | Random Forest       | [ESC-50 Dataset](https://www.karolpiczak.com/papers/Piczak2015-ESC-Dataset.pdf)              |
> > > >
> > > > ---
> > > >
> > > > ## 4.  **Key Notes About dcMltR-KAN**
> > > > 1. **Accuracy**:  Achieved **97.15% (d-index: 1.9806) **, outperforming several SOTA models.
> > > >
> > > > 2. **Classifier**:   Uses a **3-NN** classifier, which stands out compared to other SSL models relying on dense or transformer-based classifiers.
> > > >
> > > > 3. **Learning Type**:    **SSL**, effectively leveraging unlabeled data for robust environmental sound classification.
> > > > 4. Our model's performance can improve with enhanced feature extraction and parameter fine-tuning
> > > >
> > > > **Again, we APPRECIATE your insightful review!**

---

> > > > > ### Comment · Reviewer_YcJQ · 2024-12-02
> > > > >
> > > > > Dear Authors,
> > > > >
> > > > > Thank you for performing additional experiments! I find the results, in particular in respect to the number of trained parameters, impressive.
> > > > >
> > > > > Could you upload the code to replicate the additional experiments too?

---

> > > ### Author Response · Authors · 2024-12-02
> > > **Reply to reviewer YcJQ**
> > >
> > > ### Dear Reviewer,
> > >
> > > We thank you for your kind feedback.
> > >
> > > **It seems that we are unable to upload our ESC-50 and DESED codes to the ICLR2025 website after November 27, 2024 according to ICLR2025 policy**.
> > >
> > > **Our supplemental materials already include the codes for our `dcMltR-KAN` model for gravitational wave data (O1). While the new codes have some variations compared to the original ones in the supplemental materials, their overall structure remains the same.**
> > >
> > > **We also attempted to upload snapshots of two folders containing the new datasets and codes; however, the comment window does not allow such uploads. Additionally, we tried creating a Google Drive link for you, but it requires a phone number, which might violate the anonymity policy of ICLR2025.**
> > >
> > > If the paper is accepted, we should be able to upload the new datasets and codes in the supplemental materials.
> > >
> > > For your reference, we are including a README file for the ESC-50 data codes below. (DESED data codes have the same structure)
> > >
> > > ---
> > > ### README for `dcMltR-KAN_ESC-50`
> > >
> > > #### How to Run `dcMltR-KAN_ESC-50.ipynb`
> > >
> > > 1. Clone or download this repository.
> > > 2. Open the Jupyter notebook `dcMltR-KAN_ESC-50.ipynb` located in the project directory.
> > > 3. Run all cells sequentially to train and evaluate the model.
> > >
> > > ### Data
> > >
> > > Ensure that the dataset is located in the same directory as the notebook. The notebook is configured to automatically load the data from this directory.
> > >
> > > ### Notes
> > >
> > > - Ensure that your system has sufficient computational resources (a powerful GPU is HIGHLY recommended) for efficient execution of the notebook.
> > >
> > > ---
> > >
> > > **Again, we appreciate you insightful review!**

---

> > > > ### Comment · Reviewer_YcJQ · 2024-12-03
> > > >
> > > > Dear Authors,
> > > >
> > > > You could use Anonymous GitHub to share the additional experiments: https://anonymous.4open.science/
> > > >
> > > > As stated in my initial review, you presented your contribution in the niche of gravitational wave glitch detection. In my humble opinion, your paper for ICLR should be written around the novel algorithm and the different experiments should be used to support your innovation.  I acknowledge the additional experiments on ESC50 and DESED, still I score your work in the current form which does not include these experiments in the main paper or in the supplied code. This is why I decided to keep my score.
> > > >
> > > > I encourage the authors to continue with their promising approach, I am willing to vote in favour of acceptance once the ESC50 results are verifiable and presented in the main text.

---

> > > > > ### Author Response · Authors · 2024-12-04
> > > > > **Reply to Dear YcJQ reviewer**
> > > > >
> > > > > **Dear Reviewer,**
> > > > >
> > > > > **Thank you so much for your thoughtful review and valuable suggestions. We truly appreciate the time and effort you have dedicated to improving our work!**
> > > > >
> > > > > To address your comments, we have conducted additional experiments and created an anonymous GitHub repository to share the results with you:
> > > > >
> > > > > [https://anonymous.4open.science/r/testsound123-23CB/](https://anonymous.4open.science/r/testsound123-23CB/)
> > > > >
> > > > > **It has been our pleasure to work with you, and we are deeply grateful for your insightful feedback, as well as that of the other reviewers.**
> > > > >
> > > > > **Thank you again for your kindness and support!**
> > > > >
> > > > > Best regards,
> > > > > Authors

---

### Official Review · Reviewer_J537 · 2024-11-03

**Soundness:** 2
**Presentation:** 3
**Contribution:** 2
**Rating:** 3
**Confidence:** 4

**Summary:**

This paper applies debiased contrastive learning (DCL) on wave gravitational data for the task of wave glitch detection. The authors add an auxiliary Wassterstein DCL loss to address data imbalance. Additionnally, they refine the feature representations by weighting using the similarity embedding matrix. Finally, they introduce Multi-Resolution Kolmogorov-Arnold Network (KAN) layers on top of a CNN encoder to capture multi-scale patterns.

**Strengths:**

1. The paper is clearly written and easy to follow.

2. The application of contrastive learning for wave glitch detection is interesting, and is promising for extending to large unlabelled dataset.

3. The use of False Negative Elimination (FNE) at multiple levels with the multi-resolution KAN is an interesting idea.

4. The ablation studies effectively demonstrate the importance of different model components.

**Weaknesses:**

1. Unsupported claims


1.1. The authors make strong claims about KAN layers enhancing explainability, efficiency, and generalization without providing supporting evidence. The KAN layers, in fact, introduce additional complexity, and the paper does not demonstrate the claimed explainability.

1.2. Propositions and theorems are stated without proof or detailed explanation. These could have been included in an appendix for clarity and rigor.

2. Insufficient Literature Review

2.1. The literature review lacks coverage of class imbalance approaches, especially relevant methods in self-supervised contrastive learning.

2.2. Comparisons are limited to the authors' own baselines; there is no evaluation against established methods from the literature ( including on the EmoDB audio dataset).

3. The proposed approach is complex and appears unsuitable for scaling to large datasets, potentially limiting its practical applicability for large-scale learning on unlabeled data.

**Questions:**

In Section 3.1, please define L_wass for clarity. Figure 1 is very small; consider increasing its size to enhance readability. The legends in Figure 2 are very small and too difficult to read.

1. For L_wdcl (Eq. 2), why not use a single hyperparameter to balance the two losses, allowing one to scale relative to the other instead of using two hyperparameters?

2. I don't see the point of comparing two losses computed on different examples in Proposition 1, can you clarify its purpose? (also true for Theorem 3, and probably Theorem 2 too). Additionally, could you define the Rademacher complexity to improve clarity for readers unfamiliar with this concept?

---

> ### Author Response · Authors · 2024-11-26
> **Reply to Reviewer J537: Addressing Concerns and Suggestions with Gratitude**
>
> # Response
>
> ## 1.1 Explainability in KAN Layers
>
> The KAN architecture is designed for explainability, leveraging its additive structure, inspired by Kolmogorov's superposition theorem, and wavelet-based multi-resolution analysis
>
> - **Fig. S1** in the Appendix illustrates the explainability enhancement process within the KAN architecture for the SNR feature extracted from the O1 data (lines 335–336, also Section 9 in Appendix).
>
> - **Figures S2 and S3** in the Appendix demonstrate the efficiency of the KAN architecture under hierarchical loss.
>
> ---
>
> ## 1.2
> 1. All theorems have rigorous proofs provided in the Appendix (see Sections 2, 3, and 4).
>
> 2. Rigorous proofs for Propositions 1, 2, and 3 are included in the Appendix (Sections 6, 7, and 8).
>
> 3. Explanations for Theorems 1 and 2, as well as Propositions 1, 2, and 3, are provided in the updated manuscript. Theorem 3 already contains sufficient explanation.
>
> ---
>
> ## 2.1
> We respectfully disagree with this comment. To the best of our knowledge, no existing self-supervised contrastive learning or SSL methods specifically address the challenge of handling imbalanced gravitational wave data. Our paper is the first to tackle this issue.
>
> ---
>
> ## 2.2
> To the best of our knowledge, no existing methods comprehensively address all O1, O2, and O3 datasets. The baselines used in our comparisons are widely adopted for glitch detection but may not simultaneously apply to O1, O2, and O3. While one SSL method was proposed previously (lines 114–117), it showed poor performance compared to fully supervised models.
>
>
> - Additionally, three new SOTA SSL baselines for time-series data have been added.
>
> - For the EmoDB dataset, we included comparisons with SOTA methods. Refer to lines 516–519 in the updated manuscript (originally mentioned in lines 462–464), where two SOTA methods are explicitly discussed.
>
> ---
>
> ## 3
>
> We appreciate the reviewer’s comments and respectfully disagree for the following reasons:
>
> - Our method was successfully applied to the gravitational wave O1, O2, and O3 datasets, containing 41,717, 134,372, and 500,524 samples, respectively.
>
> - The impact of dcMltR-KAN on data representation (lines 484–490) and Fig. 4 illustrate the model’s effectiveness on O1, O2, and O3 data.
>
>
> - It outperformed all peer methods, including SOTA SSL approaches CPC, TS-TCC, and SimCLR (Fig. 3). CPC and TS-TCC target time-series data, while SimCLR adapts well despite not being designed for it. These results strongly confirm our model's scalability and practical value, countering claims of limited scalability.
>
>
> - We disagree with the notion that complexity undermines "practical applicability." The superior performance of our model, demonstrated against eight peer methods, validates its effectiveness and scalability, particularly for the large O3 dataset (500,524 samples).
>
> - Our ablation study also shows that simpler models fail to achieve comparable performance (see Table 1), reinforcing the necessity and robustness of our approach .
>
> ---
>
> ## Questions
>
> ### Lwass Update
> We updated the definition of $L_{\text{wass}}$ in Section 3.1 for clarity (also see Section 1 in Appendix).
>
> ### Figures 1 and 2
> We increased the size of Figures 1 and 2 to enhance readability.
>
> ---
>
> ### Q1
> The decision to use two hyperparameters, $\lambda$ and $\beta$, in $L_{\text{wdcl}}$ (Eq. 2) was made to provide finer control over the contributions of the Wasserstein loss $L_{\text{wass}}$ and the N-pair contrastive loss $L_{\text{N-pair}}$, as they address distinct objectives:
>
> - **$L_{\text{wass}}$:** Captures geometric structure and ensures robustness to data imbalance.
> - **$L_{\text{N-pair}}$:** Optimizes contrastive learning by distinguishing between positive and negative pairs.
>
> Using two hyperparameters allows independent tuning to avoid under- or over-emphasizing either term, offering flexibility for datasets with varying imbalance or complexity. A single hyperparameter would reduce this adaptability, potentially under-optimizing one loss.
>
> ---
>
> ### Q2
> The comparisons of losses in Proposition 1, Theorem 2, and Theorem 3 evaluate the impact of our techniques on training and generalization:
>
> 1. **Proposition 1**: Shows that False Negative Elimination (FNE) reduces the contrastive loss $\(L_{\text{wdcl}}\)$ by removing misleading negatives, improving embedding quality during training.
>
>
> 2. **Theorems 2 and 3**: Compare generalization error and Rademacher complexity across models with different basis functions (e.g., wavelets, B-splines) and MLPs, highlighting our model’s superior generalization and balance between complexity and performance.
>
> While comparing losses on different samples might seem unintuitive, it provides strong theoretical insights into how FNE, SBW, and multi-resolution KAN enhance optimization and generalization.
>
> We have included the norm-based Rademacher complexity definition in the updated manuscript (lines 360–364).

---

> > ### Comment · Reviewer_J537 · 2024-11-27
> >
> > Dear authors, please point me to where I can find the Appendix. I don't see it in the original pdf document.

---

> > > ### Author Response · Authors · 2024-11-27
> > > **Reply to Reviewer J537**
> > >
> > > Dear Reviewer,
> > >
> > > We apologize for any inconvenience caused. The appendix, initially included in the supplemental material, has now been appended to the end of the original manuscript for your review.
> > >
> > > Thank you for your understanding, and we appreciate your time and consideration.
> > >
> > > Best regards,
> > >
> > > Authors

---

> ### Comment · Reviewer_J537 · 2024-11-29
>
> Dear authors,
>
> Thank you for addressing my concerns. Regarding point 2.1, I was referring to works in the contrastive learning literature that specifically address data imbalance, rather than those focused solely on wave glitch detection.
>
> To clarify, here are some examples:
>
> In [1], the authors apply contrastive learning on two branches, before and after pruning. They assume that tail samples are more easily "forgotten" by the pruned model and propose forcing these samples to be remembered by the pruned model.
>
> In [2], the approach leverages the loss value for each input to identify tail samples, applying stronger augmentations to those samples to enhance their representation.
>
> In [3], they oscillate the temperature of the InfoNCE loss to alternate between instance discrimination and group discrimination. This ensures the model learns both group-level features and instance-specific details.
>
> Since one of your primary motivations is robustness to data imbalance, the lack of related work in this area and the absence of comparisons with other methods addressing this challenge make it difficult to quantify the advantage of your approach.
>
>
> [1] Z. Jiang et al. Self-damaging contrastive learning, ICML 2021
>
> [2] Z. Zhou et al. Contrastive learning with boosted memorization, ICML 2022
>
> [3] A. Kukleva et al. Temperature Schedules for self-supervised contrastive methods on long-tail data, ICLR 2023

---

> > ### Author Response · Authors · 2024-11-29
> > **Reply to Reviewer J537**
> >
> > # Response
> >
> > Thank you for clarifying your concern and for providing examples from the contrastive learning literature that address data imbalance. The methods you referenced ([1], [2], and [3]) offer innovative strategies, such as leveraging pruning ([1]), augmentations for tail samples ([2]), and temperature oscillations to balance instance- and group-level learning ([3]).
> >
> > We would like to address your points as follows:
> >
> > ## 1.
> > Our work focuses specifically on gravitational wave glitch detection, which presents unique challenges such as overlapping noise, rare glitch classes, and the need to preserve astrophysical relevance. **As a result, our literature review exclusively engages with works specific to gravitational wave data and its associated machine learning models**. Unlike the datasets addressed in [1], [2], and [3], gravitational wave datasets require domain-specific methods to handle their unique characteristics.
> >
> > We are not attempting to address the broader data imbalance challenges discussed in [1], [2], and [3] in this study. Instead, we propose a tailored framework to address data imbalance exclusively within the context of gravitational wave glitch detection. This framework includes:
> >
> > - **Wasserstein-Based Loss**: Aligns majority and minority sample distributions by accounting for the geometry of the data.
> > - **False Negative Elimination (FNE)**: Reduces noise by removing mislabeled negatives, improving representation quality.
> > - **Similarity-Based Weighting (SBW)**: Amplifies the influence of rare samples, enhancing the representation of minority classes.
> > - **MltR-KAN**: Captures both global and local structures using multi-resolution analysis with wavelets, ensuring minority class features are preserved.
> >
> > These methods are tailored to astrophysical data and may not directly apply to more general contexts like those in [1], [2], and [3].
> >
> > ---
> >
> > ## 2.
> > **Our framework has been rigorously tested on ALL currently available gravitational wave datasets: O1, O2, and O3, where O3 has more than half million observations.**  **Unlike benchmark datasets in AI, the scope of gravitational wave data is limited, with only these three datasets available for this task**. The upcoming O4 dataset, expected in 2025, will provide an opportunity for further validation.
> >
> > **Our results strongly demonstrate the effectiveness of our method in addressing data imbalance within this exclusive context.** Key benefits include:
> >
> > - **Improved Representation for Rare Glitches**: SBW and FNE work together to preserve minority class samples.
> > - **Noise Mitigation**: FNE ensures the model learns from accurate relationships by removing mislabeled negatives.
> > - **Multi-Resolution Feature Analysis**: MltR-KAN balances local and global data structures, enabling robust embeddings and better generalization.
> >
> > Furthermore, **Theorem 1** provides a theoretical guarantee of the robustness of our Wasserstein-based Debiased Contrastive Learning (wDCL) to data imbalance from a loss-function perspective. This result is more general and not limited to gravitational wave data, offering insights into the potential applicability of wDCL for other domains dealing with data imbalance.
> >
> > ---
> >
> > ## 3.
> > We recognize the relevance of broader contrastive learning methods such as [1], [2], and [3]. However, our focus on gravitational wave data required us to prioritize domain-specific solutions. Due to time constraints, we were unable to include direct comparisons with these methods in this manuscript.
> >
> > **If the paper is accepted, we plan to:
> > - Include a discussion of these methods and their relevance to our contributions in the literature review section.
> > - Explicitly propose comparisons with these methods as part of our future work section, providing a roadmap for extending our research to broader datasets and challenges.**
> >
> > ---
> >
> > ## 4.
> > **We emphasize that our work demonstrates significant results in addressing data imbalance in gravitational wave glitch detection. This is strongly evidenced by our model’s performance on the O1, O2, and O3 datasets, where it elegantly outperformed eight peer methods.**
> >
> > While comparisons with general contrastive learning methods such as [1], [2], and [3] are valuable, **our work is firmly rooted in the astrophysical domain, addressing its unique challenges with tailored methods.** We are confident in the applicability of our approach to all currently available gravitational wave datasets and look forward to further validating it with the O4 dataset in the future.
> >
> > ---
> >
> > ## 5.
> > We appreciate your thoughtful feedback and hope our detailed response clarifies our focus on gravitational wave glitch detection and its exclusive literature. Our contributions are specifically designed for astrophysical data, where unique challenges like overlapping noise and rare glitch classes require innovative solutions.
> >
> > **Again, we appreciate all your insightful comments so much!**

---

### Official Review · Reviewer_3aUm · 2024-11-06

**Soundness:** 3
**Presentation:** 3
**Contribution:** 3
**Rating:** 6
**Confidence:** 3

**Summary:**

The paper presents a new KAN-based model to solve time-series gravitational wave glitch detection problem. It is characterized by its multi-resolution KAN architecture and debiased contrastive learning loss that improves the CNN-based encoder. The paper presents experimental results on the benchmark datasets (also including an audio dataset), showcasing performance improvement over common supervised-learning methods.

**Strengths:**

- The paper provides rigorous mathematical definitions of their methods. Easy to follow and understand the proposed concept.
- Although the reviewer is not an expert in that particular domain area, the proposed method seems to have improved over the reasonable supervised-learning models.
- The paper did ablation tests to prove that each of the newly proposed methods is meaningful.

**Weaknesses:**

- It is not clear why Wasserstein distance needs to be used. The explanation implies the imbalanced data could be an issue by comparing the distributions, instead of Euclidean distance. But wouldn't the same argument be made via KL-divergence?

- It could be based on my lack of experience with the dataset, but the way the dataset was divided into training and testing folds, while the information on validation is missing, is not too informative. More detail is needed, as to how the model was trained.

- The construction of the similarity matrix seems to be very computationally demanding. Meanwhile, it also means that the features can be based on the original similarity defined in the CNN's output. The additional contrastive learning part does introduce some performance improvement, but it is a little difficult to wrap my head around how these two different approaches interact with each other.

**Questions:**

- In eq (2), what's the meaning of x(\alpha)? I understand what \alpha actually means, but x as a function of \alpha doesn't make sense to me.

- Is there any better way to construct the positive pair for contrastive learning? Curious if adding Gaussian noise generalizes well enough to unseen examples. For example, in text processing, positive word pairs can be constructed by using context windows. Is there any way to do something like this in this dataset?

---

> ### Author Response · Authors · 2024-11-26
> **Reply to Reviewer 3aUm : Addressing Insights and Suggestions with Gratitude**
>
> # Responses
>
> ## Weaknesses 1
> ### Wasserstein Distance vs. KL-Divergence
> Wasserstein distance measures the "cost" of transforming one distribution into another, making it effective for non-overlapping distributions by accounting for data geometry. In contrast, KL-divergence assumes overlapping supports and diverges when distributions do not overlap, limiting its stability in imbalanced data.
>
> ### Sensitivity to Outliers
> KL-divergence focuses on relative entropy and is highly sensitive to small probabilities, often exaggerating noise or outliers. Wasserstein distance, by considering geometric structure, provides a more comprehensive and robust metric for high-dimensional or imbalanced data.
>
> ### Metric Properties
> Unlike KL-divergence, Wasserstein distance is a true metric with symmetry, ensuring unbiased and consistent comparisons. Its symmetry avoids directional bias, making it a more reliable choice for analyzing imbalanced distributions.
>
> ---
>
> ## Weaknesses 2
> The dataset division into training and testing folds was designed for evaluating fully-supervised baseline models, focusing on test set performance without requiring an explicit evaluation set. Hyperparameter tuning or model selection is not the primary goal for these comparisons.
>
> For the proposed SSL model, dcMltR-KAN, evaluation differs. We assess representation quality using top-1 accuracy of a k-nearest neighbors (kNN) classifier, which does not require a traditional training/testing split. This approach, common in SSL literature, emphasizes representation quality over downstream performance. These points have been clarified in the updated manuscript.
>
> ---
>
> ## Weaknesses 3
> The similarity matrix is expensive for large datasets like O3 (500,524 observations) but can be mitigated with GPU acceleration, sparsification, mini-batching, and efficient libraries like cuML (see lines 523–529).
>
> ---
>
> ## Explanation of CNN-Based Feature Extraction and Contrastive Learning Framework
>
> 1. **CNN as Feature Encoder**
>    The CNN extracts high-level features, forming the foundation for downstream representation learning in the contrastive framework.
>
> 2. **Similarity Matrix and Contrastive Learning**
>    The similarity matrix, built from CNN features, quantifies pairwise relationships and enables refinement via:
>    - **False Negative Elimination (FNE):** Removes mislabeled negatives, reducing noise.
>    - **Similarity-Based Weighting (SBW):** Assigns weights to pairs based on similarity scores, emphasizing relevant pairs and refining the embedding space.
>
> 3. **Wasserstein-Loss-Based Contrastive Learning**
>    The CNN-derived features, refined through FNE and SBW, are further optimized using a Wasserstein loss. This explicitly pushes similar samples closer together while separating dissimilar ones, creating a more discriminative embedding space.
>
> 4. **mltR-KAN Refinement**
>    mltR-KAN enhances embeddings by capturing global and local structures through a multi-resolution approach, amplifying contrastive learning benefits. Combined with FNE, SBW, and Wasserstein loss, it creates hierarchical embeddings that are robust to noise, interpretable, and ideal for large-scale datasets like O3.
>
> ---
>
> ## Q1
> In Equation (2), $x(\alpha)\$ represents the input \(x\) modulated or filtered by the hyperparameter $\alpha$, which acts as a discriminative threshold. The hyperparameter $\alpha$ determines the threshold for classifying sample pairs as positive or negative, and \(x(\alpha)\) formalizes how the input $x$ is categorized based on this threshold.
>
> - **$x(\alpha)^-\$: Negative Pairs**
>   These are pairs with similarity scores below $\alpha$, filtering out irrelevant or noisy negatives while retaining meaningful ones for contrastive learning. By focusing on hard negatives, the model sharpens embeddings, enhancing class separation.
>
> - **No $x(\alpha)^+\$:**
>   Positive samples $\(x^+\)$ are explicitly defined as true positives and do not require filtering by $\alpha$, as they inherently represent the same class, providing a clear signal for learning.
>
> ---
>
> ## Q2
> Constructing effective positive pairs for contrastive learning in gravitational wave data involves leveraging the unique characteristics of the data. Currently, adding Gaussian noise is a highly suitable approach, as it aligns with the inherently noisy nature of gravitational wave signals. This method effectively creates realistic variations, helping the model learn robust representations that generalize to unseen examples.
>
> Additional strategies include:
> - **Frequency-domain augmentations** (e.g., band-pass filtering, frequency shifts) to simulate variability.
> - **Time-domain techniques** (e.g., time warping) for timing and duration changes.
> - **Feature-space nearest neighbors** to identify similar samples dynamically.
>
> While these methods enhance diversity and robustness, they may increase computational costs and risk overfitting if augmentations deviate from realistic signal variations.

---

> > ### Author Response · Authors · 2024-12-02
> > **We appreciate your insightful review!**
> >
> > Dear Reviewer:
> >
> > We **APPRECIATE** your insightful review!
> >
> > Sincerely,
> > The Authors

---

> > > ### Comment · Reviewer_3aUm · 2024-12-03
> > > **appreciate the clarification**
> > >
> > > I appreciate the clarification. There are a couple of remaining issues though:
> > > - I understand the difference between KL-div and Wasserstein distance and that's why I asked this question; the original manuscript only mentioned W-distance as an alternative to Euclidean. Is there evidence that they model is dealing with non-overlapping distributions, necessitating W-distance?
> > > - With kNN clarification, you still need some validation, e.g., to define k.

---

> > > > ### Author Response · Authors · 2024-12-03
> > > > **Reply to Reviewer 3aUm**
> > > >
> > > > Dear Reviewer:
> > > >
> > > > Thank you for raising these insightful points.
> > > > **We appreciate the opportunity to address them further.**
> > > >
> > > > ## 1. Necessity of Wasserstein Distance for Non-Overlapping Distributions
> > > > In the original manuscript, we positioned Wasserstein distance as an alternative to Euclidean distance primarily due to its ability to account for the geometry of data distributions. Thanks to your suggestion, we have considered including a comparison with KL-divergence in our updated manuscript, which further supports the use of Wasserstein distance.
> > > >
> > > > **Evidence for Non-Overlapping Distributions in Our Model**:
> > > > - **Imbalanced Data Characteristics**: In gravitational wave glitch detection, rare glitch classes often exhibit sparse and scattered distributions in feature space compared to majority classes. These patterns result in distributions with minimal overlap, especially for minority classes.
> > > > - **Theoretical Guarantee**: Theorem 1 in our manuscript provides a formal guarantee of the robustness of the Wasserstein-based Debiased Contrastive Loss (wDCL) to imbalanced distributions. This theoretical foundation reinforces the suitability of Wasserstein distance, even in cases of minimal distribution overlap.
> > > > - **Embedding Analysis**: Post-embedding visualization using tools like UMAP shows distinct, often disjoint clusters for minority classes, supporting the need for a distance metric that can effectively handle non-overlapping distributions.
> > > >
> > > > Wasserstein distance's ability to align majority and minority class distributions by modeling the "transport cost" between them proves valuable, particularly when distributions are imbalanced or exhibit disjoint supports.
> > > >
> > > > ## 2. Validation for Selecting \( k \) in kNN
> > > > You raise an important point about the necessity of validating the choice of \( k \) in kNN for evaluation. Here’s how we approached and justified our choice:
> > > >
> > > >
> > > > - **General Use of 1-NN Prediction**:
> > > >   Generally, we use 1-NN prediction as most SSL models do, meaning \( k = 1 \). 1-NN prediction is beneficial because it directly evaluates the quality of learned representations by testing whether the nearest neighbor in the feature space belongs to the same class. It is a simple, non-parametric approach that does not introduce additional hyperparameters, making it effective for assessing how well the model clusters similar instances. Moreover, it avoids overfitting to local noise, which can happen with larger values of \( k \). This makes 1-NN particularly suitable for self-supervised learning (SSL), where the primary goal is to evaluate feature embeddings.
> > > >
> > > > - **Special Use of 3-NN Prediction**:
> > > >   In specific cases, such as noisy data, limited sample sizes, ambiguous classes, or high variability between classes, we use \( k = 3 \), 3-NN for prediction. Under such conditions, 1-NN prediction can be biased, while 3-NN provides more robust results, improved generalization, and better alignment with practical applications. Among the six datasets tested with our model, we use 3-NN prediction for ESC-50 and 1-NN for all the other datasets.
> > > >
> > > >
> > > > **Again, thank you for your insightful review. We appreciate your kind feedback immensely!**

---

### Note · Authors · 2025-03-01

I have read and agree with the venue's withdrawal policy on behalf of myself and my co-authors.

---

### Meta-Review · Area_Chair_VZkG · 2024-12-16

**Metareview:**

The work presents a KAN-based model designed to tackle the problem of time-series gravitational wave glitch detection. It features a multi-resolution KAN architecture and incorporates a debiased contrastive learning loss to improve the performance of its CNN-based encoder. Four reviewers evaluated this work, providing very useful comments and questions. At the end of the discussion phase, one of the reviewer was timidly positive about the work, considering it marginally above the acceptance threshold. The remaining three reviewers were less positive. In particular, two of the reviewers are slightly against accepting the work; the remaining reviewer considers the work is not ready for publication.


The key claims are: (i) the proposal of a novel self-supervised learning (SSL) approach that enhances glitch detection, robustness, explainability, and generalization, and (ii) the introduction of three  key novel components: Wasserstein Debiased Contrastive Learning (wDCL), a CNN-based encoder, and a Multi-Resolution KAN. The stength of the paper resides mainly in the application of contrastive learning for wave glitch detection, the  use of False Negative Elimination (FNE) at multiple levels with a multi-resolution KAN.

Chief concerns regard: (i) lack of explanation of some technical choices, namely how k was chosen in kNN,  and the need of Wasserstein Distance for Non-Overlapping Distributions, (ii) limited experimental validation, and  (iii) positioning of the paper.

The authors adequately addressed their choice of the Wasserstein distance during the discussion phase; however, their justification for the choice of  k, particularly when k=3, appeared less well-founded. The initial limited experimental validation was expanded with additional experiments on two datasets to demonstrate generalizability across different tasks, namely they used sound event detection. Notably, Reviewer YcJQ viewed these new results positively, while Reviewer jin5 felt that the performance gap with the SOTA model remains too large to firmly support claims of generalizability. With respect to the latter concern (iii),  it seems the authors have not fully resolved it, even in the discussion phase. Specifically, the work lacks a comprehensive literature review and a meaningful comparison with other well-established, more general approaches to addressing imbalance. Consequently, while the contribution is interesting, it remains somewhat isolated and does not convincingly situate itself within the broader context of existing research.

**Additional Comments On Reviewer Discussion:**

The discussion phase was highly engaging, with all reviewers actively participating. However, I must point out that the authors were somewhat out of line in accusing Reviewer jin5 of unprofessionalism, despite the reviewer providing sound technical feedback and raising valid points. Additionally, the reviewer’s overall recommendation was not strongly negative, and their concerns about the model's generalizability are not entirely unwarranted.

During the discussion, the authors effectively addressed most of the technical comments raised by the four reviewers and provided further clarification on additional questions that emerged. Nevertheless, the work, while interesting and promising, still lacks a thorough literature review and a meaningful comparison with other well-established, more general approaches to addressing imbalance. Moreover, the claims regarding generalizability may require further validation.

---

### Decision · Program_Chairs · 2025-01-22

Reject